# Adversarial Resilience in Sequential Prediction via Abstention

**Surbhi Goel**
University of Pennsylvania
surbhig@cis.upenn.edu

**Steve Hanneke**
Purdue University
steve.hanneke@gmail.com

**Shay Moran**
Technion and Google Research, Israel
smoran@technion.ac.il

**Abhishek Shetty**
University of California, Berkeley
shetty@berkeley.edu

## Abstract

We study the problem of sequential prediction in the stochastic setting with an adversary that is allowed to inject clean-label adversarial (or out-of-distribution) examples. Algorithms designed to handle purely stochastic data tend to fail in the presence of such adversarial examples, often leading to erroneous predictions. This is undesirable in many high-stakes applications such as medical recommendations, where abstaining from predictions on adversarial examples is preferable to misclassification. On the other hand, assuming fully adversarial data leads to very pessimistic bounds that are often vacuous in practice.

To move away from these pessimistic guarantees, we propose a new model of sequential prediction that sits between the purely stochastic and fully adversarial settings by allowing the learner to abstain from making a prediction at no cost on adversarial examples, thereby asking the learner to make predictions with certainty. Assuming access to the marginal distribution on the non-adversarial examples, we design a learner whose error scales with the VC dimension (mirroring the stochastic setting) of the hypothesis class, as opposed to the Littlestone dimension which characterizes the fully adversarial setting. Furthermore, we design learners for VC dimension 1 classes and the class of axis-aligned rectangles, which work even in the absence of access to the marginal distribution. Our key technical contribution is a novel measure for quantifying uncertainty for learning VC classes, which may be of independent interest.

## 1 Introduction

Consider the problem of sequential prediction in the realizable setting, where labels are generated from an unknown $f^*$ belonging to a hypothesis class $\mathcal{F}$. Sequential prediction is typically studied under two distributional assumptions on the input data: the stochastic setting where the data is assumed to be identically and independently distributed (i.i.d) according to some fixed (perhaps unknown) distribution, and the fully-adversarial setting where we make absolutely no assumptions on the data generation process. A simple empirical risk minimzation (ERM) strategy works for the stochastic setting where the learner predicts according to the best hypothesis on the data seen so far. The number of mistakes of this strategy typically scales with the Vapnik-Chervonenkis (VC) dimension of the underlying hypothesis class $\mathcal{F}$. However, in the fully adversarial setting, this strategy can lead to infinite mistakes even for classes of VC dimension 1 even if the adversary is required to be consistent with labels from $f^*$. The Littlestone dimension, which characterizes the complexity of sequential prediction in fully-adversarial setting, can be very large and often unbounded compared to

the VC dimension [Lit87]. This mismatch has led to the exploration of beyond worst-case analysis for sequential prediction [RST11, HRS20, RS13a, BCKP20].

In this work, we propose a new framework that sits in between the stochastic and fully-adversarial setting. In particular, we consider sequential prediction with an adversary that injects adversarial (or out-of-distribution) examples in a stream of i.i.d. examples, and a learner that is allowed to abstain from predicting on adversarial examples. A natural motivation for our framework arises in medical diagnosis where the goal is to predict a patient's illness based on symptoms. In cases where the symptoms are not among the commonly indicative ones for the specific disease, or the symptoms may suggest a disease that is outside the scope of the doctor's knowledge, it is safer for the doctor to abstain from making a prediction rather than risk making an incorrect one. Similarly, for self-driving cars, in cases where the car encounters weather conditions outside of its training, or unknown information signs, it is better for the algorithm to hand over access to the driver instead of making a wrong decision which could end up being fatal.

In the proposed framework, the learner's goal is to minimize erroneous predictions on examples that the learner chooses to predict on (i.i.d. or adversarial) while refraining from abstaining on too many i.i.d. examples. If the learner was required to predict on every example, then the adversary could produce a fully-adversarial sequence of examples which would force the learner to make many erroneous predictions. The abstention option allows us to circumvent this challenge and handle any number of adversarial injections without incurring error proportional to the number of injections. In this framework, we can ask the following natural question:

*Is there a statistical price for certainty in sequential prediction?*

In particular, can we recover stochastic-like guarantees in the presence of an adversary if we are allowed to abstain from predicting on adversarial examples? A priori, it is not clear where on the spectrum between the fully-adversarial and stochastic models, the complexity of this problem lies. The main challenges arise from the fact that the adversary fully controls the injection levels and provides no feedback about which examples were adversarial, and the learner has to perform one-sample outlier detection, which is nearly impossible. Despite this, we show that it is possible to guarantee certainty in a statistically efficient manner.

## 1.1 Main Contributions

We summarize the main contributions of our work:
- We formalize a new framework of beyond-worst case learning which captures sequential prediction on a stochastic sequence with a clean-label injection-only adversary. With the option of abstention, our framework allows for any number of injections by the adversary without incurring error proportional to the number of injections. Our notion of error simultaneously guarantees few mistakes on classified data while ensuring low abstention rate on non-adversarial data. Our framework naturally connects to uncertainty quantification and testable learning.
- In our framework, we design an algorithm that achieves error $O(d^2 \log T)$ for classes with VC dimension $d$ for time horizon $T$, given access to the marginal distribution over the i.i.d. examples. This allows us to get (up to a factor of $d$) the guarantees of the stochastic setting while allowing for any number of adversarial injections.
- We further design an algorithm that achieves $O(\sqrt{T \log T})$ error for the special (but important) case of VC dimension 1 classes without any access to the marginal distribution over the i.i.d. examples. Similar ideas also allow us to design an algorithm for the class of axis-aligned rectangles in any dimension $d$ with error $O(d\sqrt{T \log T})$.

Our algorithms uses a novel measure of uncertainty for VC classes to identify regions of high uncertainty (where the learner abstains) or high information gain (where the learner predicts and learn from their mistakes). The measure uses structural properties of VC classes, in particular, shattered sets of varying sizes. In the known distribution setting, our measure is easy to compute, however for the unknown distribution setting, we show how to design a proxy using only the examples we have seen so far using a leave-one-out type strategy.

## 1.2 Related Work

**Beyond-worst case sequential prediction.** Due to pessimistic nature of bounds in adversarial online learning, there are several frameworks designed to address this issue. One approach is to consider

mild restrictions on the adversarial instances such as slight perturbation by noise. This has been formalized as the smoothed adversary model (see [RST11, HRS20, HRS22, HHSY22, BDGR22]) and has been used to get statistical and computationally efficient algorithms. Another approach has been to make the future sequences more predictable given the past instances. Examples of such settings are predictable sequence [RS13b], online learning with hints [BCKP20], and notions of adaptive regret bounds [FRS20].

**Abstention-based learning.** Abstention has been considered in several other works in classification, both in the online and offline settings. An early example of this is the Chow reject model [Cho70]. Various versions of this have been considered in offline learning (see e.g. [HW06, BW08, BZ20] and references therein) and online learning (see e.g. [ZC16, CDG$^+$19, NZ20] and references therein). These results show that abstention can lead to algorithms with desired features, for example fast rates without margins assumptions. Another line of work that is closely related to our setting is the KWIK (*knows what it knows*) framework by [LLW08] which requires the learner to make predictions only when it is absolutely confident, and abstain otherwise. This requirement was relaxed to allow for mistakes by [SZB10]. The key difference from our work is that they assume a fully-adversarial stream thus the error bounds can be as large as the size of the domain, which is unbounded in the settings we consider. Perhaps, the work that is closest to our setting is the study of adversarial learning with clean-label injections by [GKKM20, KK21]. In their *transductive* adversarial model, the learner is given labeled training examples and unlabeled test examples on which it must predict, where the test examples may have been injected by an adversary. They show how to abstain with few test misclassifications and few false abstentions on non-adversarial examples. However, in many real-world scenarios, it is unrealistic to expect to have the entire test set in advance, which further motivates the fully online setting that we consider.

**Adversarially robust learning.** Highly related to our setting is the problem of *inductive* learning in the presence of adversarial corruptions. The literature on this is generally divided into two scenarios: test-time attacks and training-time attacks. In the case of test-time attacks, the learning algorithm is trained on an (uncorrupted) i.i.d. training set, but its *test* examples may be corrupted by an adversary whose intention is to change the classification by corrupting the test example [SZS$^+$13, BCM$^+$13, GSS15, FMS18, AKM19, MHS19, MHS20, MHS21, MHS22, MGDS20]. Often the goal in this setting is to learn a classifier that predicts correctly on all adversarial test examples, which is a very strong requirement. Empirical work in this space has focused on designing methods to make training adversarially robust [MMS$^+$17, WK18], and also on detecting adversarial examples [PDDZ18, AHFD22]. Detecting adversarial examples is a very challenging tasks and proposed solutions are often brittle [CW17]. In fact, our framework does not explicitly require detection as long as we can predict correctly on these. On the other hand, in the case of training-time attacks, the training data the learning algorithm trains on is corrupted by an adversary (subject to some constraints on what fraction it may corrupt and what types of corruptions are possible), while the test examples are uncorrupted [Val85, KL93, BEK02, BNL12, ABL17, SKL17, SHN$^+$18, LF21, GKM21, HKM$^+$22, BBHS22]. In particular, within this literature, most relevant to the present work is the work on *clean-label* poisoning, where the adversary's corrupted examples are still labeled correctly by the target concept [SHN$^+$18, BHQS21]. Comparing to these works, it is interesting to note that the fact that our setting involves *sequential* prediction (i.e., online), our problem may be viewed simultaneously as *both* training-time and test-time corruption: that is, because on each round the point we are predicting (or abstaining) on may be inserted by an adversary, this could be viewed as a test-time attack; on the other hand, since the prefix of labeled examples we use to make this prediction may also contain adversarial examples, this can also be viewed as a training-time attack. Thus, our setting requires reasoning accounting for issues arising from both attack scenarios, representing a natural blending of the two types of scenarios.

## 2 Abstention Framework

In this section, we present the formal framework for sequential prediction with abstentions. For background on sequential prediction and other related concepts, please refer to Appendix A.

**Notation.** We will denote the domain with $\mathcal{X}$ and the distribution over $\mathcal{X}$ as $\mathfrak{D}$. We let $\Delta(\mathcal{X})$ denote the set of all distributions over $\mathcal{X}$. We will work in the realizable setting where our label will be according to some function in $\mathcal{F}$ Given a class $\mathcal{F}$ and a data set $S = \{(x_i, y_i)\}$, we will denote

by $\mathcal{F}|_S$, the class $\mathcal{F}|_S = \{f \in \mathcal{F} : \forall i \quad f(x_i) = y_i\}$. When the data set contains a single point $S = \{(x, y)\}$, it will be convenient to denote $\mathcal{F}|_S$ as $\mathcal{F}^{x \to y}$.

**Protocol.** At the start, the adversary (or nature) picks a distribution $\mathfrak{D}$ over the domain $\mathcal{X}$ and the labelling function $f^\star \in \mathcal{F}$. We will be interested in both the setting where the learner knows the distribution $\mathfrak{D}$ and the setting where the learner does not know the distribution $\mathfrak{D}$. In the traditional sequential prediction framework, the learner sees input $x_t$ at time $t$ and makes a prediction $\hat{y}_t$ and observes the true label $y_t$. The main departure of our setting from this is that an adversary also decides before any round whether to inject an arbitrary element of $\mathcal{X}$ (without seeing $x_t$). We denote by $\hat{x}_t$ the instance after the adversarial injection ($\hat{x}_t = x_t$ or $\hat{x}_t \neq x_t$). The learner then observes $\hat{x}_t$ and makes a prediction $\hat{y}_t$ and observes the true label $y_t$, as in the traditional sequential prediction framework. We present this formally below.

---

**Protocol 1** Sequential Prediction with Adversarial Injections and Abstentions

---

Adversary (or nature) initially selects distribution $\mathfrak{D} \in \Delta(\mathcal{X})$ and $f^\star \in \mathcal{F}$. The learner does not have access to $f^\star$. The learner may or may not have access to $\mathfrak{D}$.

**for** $t = 1, \ldots, T$ **do**
    Adversary decides whether to inject an adversarial input in this the round ($c_t = 1$) or not ($c_t = 0$).
    **if** $c_t = 1$ **then** Adversary selects any $\hat{x}_t \in \mathcal{X}$
    **else** Nature selects $x_t \sim \mathfrak{D}$, and we set $\hat{x}_t = x_t$.
    Learner receives $\hat{x}_t$ and outputs $\hat{y}_t \in \{0, 1, \bot\}$ where $\bot$ implies that the learner abstains.
    Learner receives clean label $y_t = f^\star(\hat{x}_t)$.

---

*Remark.* It is important to note we are in the realizable setting, even after the adversarial injections since the labels are always consistent with a hypothesis $f^\star \in \mathcal{F}$. This model can naturally be extended to the agnostic setting with adversarial labels. It is also possible to allow the adversary to adaptively choose $f^\star \in \mathcal{F}$, that is, the adversary has to make sure the labels are consistent with some $f \in \mathcal{F}$ at all times but does not have to commit to one fixed $f^\star$. Another interesting variation would be change the feedback to not include the true label $y_t$ on rounds that the learner abstains. As we will see, the known distribution case will be able to handle adaptive $f^\star$ and limited label access.

**Objective.** In our framework, the goal of the learner is to have low error rate on the rounds it decides to predict (that is when $\hat{y}_t \in \{0, 1\}$) while also ensuring that it does not abstain ($\hat{y}_t = \bot$) on too many non-adversarial rounds ($c_t = 0$). More formally, the learner's objective is to minimize the following error (or regret),

$$\mathsf{Error} := \underbrace{\sum_{t=1}^{T} \mathbb{1}[\hat{y}_t = 1 - f^\star(\hat{x}_t)]}_{\mathsf{MisclassificationError}} + \underbrace{\sum_{t=1}^{T} \mathbb{1}[c_t = 0 \wedge \hat{y}_t = \bot]}_{\mathsf{AbstentionError}}.$$

It is important to note that we allow the learner to abstain on adversarial examples for free. This allows the learner to have arbitrarily many injections without paying linearly for them in the error.

*Remark.* There are many natural generalizations and modification of the model and objective that one could consider. For example, we would consider a cost-based version of this objective that would allow us to trade-off these errors.

**Connections to testable learning.** A further interesting connection can be made by viewing our model as an online version of testable learning framework of [RV22]. In order to see the analogy more direct, we will focus on the setting where the learner knows the distribution $\mathfrak{D}$. In the setting, a learning algorithm is seen as a tester-learner pair. The tester takes the data set as input and outputs whether the data set passes the test. The algorithm then takes as input any data set that passes the test and outputs a hypothesis. The soundness guarantee for the pair of algorithms is that the algorithm run on any data set that passes the test must output a hypothesis that is good on the distribution. The completeness requires that when the dataset is indeed from the "nice" distribution, then the tester passes with high probability. We can see our framework in this light by noting that the decision of whether to abstain or not serves as a test. Thus, in this light, completeness corresponds to the abstention error being small when the data is non-adversarial i.e. is from the true distribution, while the soundness corresponds to the misclassification error being small on points the algorithm decides not to abstain. While the testable learning literature primarily focuses on the computational aspects of learning algorithms, our focus is solely on the statistical aspects.

# 3 Warm-up: Disagreement-based Learners

As a first example to understand the framework, we consider the most natural learner for the problem. Given the data $S$ of the examples seen thus far, the learner predicts on examples $\hat{x}$ whose labels it is certain of. That is, if there is a unique label for $\hat{x}$ consistent with $\mathcal{F}|_S$, the learner predicts that label. Else, it abstains from making a prediction. This region of uncertainty is known as the disagreement region.

**Example: thresholds in one dimension.** Consider learning a single-dimensional threshold in $[0, 1]$ (that is, concepts $x \mapsto \mathbb{1}[x \geq t]$ for any $t \in [0, 1]$). While it is well known that ERM achieves $\log T$ misclassification error for i.i.d. data sequences, in the case of an adversarially chosen sequence, it is also well known that the adversary can select inputs in the disagreement region each time (closer and closer to the decision boundary) and thereby force any non-abstaining learner to make a linear number of mistakes (recall that the Littlestone dimension of thresholds is infinite [Lit87]). Indeed, it is known that the function classes $\mathcal{F}$ for which non-abstaining predictors can be forced to have $\mathsf{MisclassificationError} = \Omega(T)$ are precisely those with embedded threshold problems of unbounded size [Lit87, She78, Hod97, ALMM19]. Let us now consider the learner that abstains in the disagreement region and predicts based on the consistent hypothesis outside of this region.

**Proposition 3.1.** *Disagreement-based learner for one dimensional thresholds has*

$$\mathsf{MisclassiciationError} = 0 \text{ and } \mathsf{AbstentionError} \leq 2 \log T.$$

To see this note that our learner only predicts when the input is not in the disagreement region and thus it never predicts incorrectly ($\mathsf{MisclassiciationError} = 0$). As for the abstentions, a simple exchangeability, argument shows that when there are $n$ i.i.d. examples in the sequence, the probability of the new non-adversarial example being in the disagreement region is $1/n$. Summing this over the time horizon gives us the above proposition.

**Perfect selective classification and active learning.** The learner for the above thresholds problem is a well-known strategy from the areas of perfect selective classification and active learning known as *disagreement-based* learning [RS88, EYW10, CAL94, BBL09, DHM07, HY15]. In the perfect selective classification setting [RS88, EYW10], the learner observes the examples sequentially, as in our framework, and may predict or abstain on each, and must satisfy the requirement that whenever it predicts its prediction is *always* correct. From this fact, it immediately follows that applying any perfect selective classification strategy in our setting, we always have $\mathsf{MisclassiciationError} = 0$, so that its performance is judged purely on its abstention rate on the iid examples. It was argued by [EYW10] that the optimal abstention rate among perfect selective classification strategies is obtained by the strategy that makes a prediction on the next example if and only if all classifiers in the hypothesis class that are correct on all examples observed so far, *agree* on the example. Note that this is precisely the same learner used above. This same strategy has also been studied in the related setting of *stream-based active learning*[1] [CAL94, BBL09, Han09, Han14, DHM07, HY21]. The abstention rate achievable by this strategy for general hypothesis classes is thoroughly understood [Han07, Han11, Han09, Han12, Han14, Han16, EYW10, EYW12, WHE15, HY15]. In particular, a complete characterization of the optimal distribution-free abstention rate of perfect selective classification is given by the *star number* of the hypothesis class [HY15, Han16]. The star number $\mathfrak{s}$ is the size of the largest number $s$ such that there are examples $\{x_1, \ldots, x_s\}$ and hypotheses $h_0, h_1, \ldots, h_s$ such that $h_i$ and $h_0$ disagree exactly on $x_i$. For instance, $\mathfrak{s} = 2$ for threshold classifiers [HY15]. It was shown by [Han16] that the optimal distribution-free abstention rate for perfect selective classification is sublinear if and only if the star number is finite (in which case it is always at most $\mathfrak{s} \log T$). One can show that $\mathfrak{s}$ is always lower bounded by the VC dimension of the class. Unfortunately, the star number is infinite for most hypothesis classes of interest, even including simple VC classes such as *interval* classifiers [HY15].

**Beyond disagreement-based learning.** The learner that abstains whenever it sees an example that it is not certain of may be too conservative. Furthermore, it does not exploit the possibility of learning from mistakes. Let us consider another example to elucidate this failure. Consider the

---

[1]In this setting, instead of observing a sequence of labelled examples, the learner only observes the examples without their target labels, and at each time may query to observe the target label. The disagreement-based strategy chooses to query precisely on the points for which the classifiers in the hypothesis class correct on the observed labels so far do not all agree on the label [CAL94]. The rate of querying for this strategy is precisely the same as the abstention rate in the perfect selection classification setting [Han11, Han14, EYW12].

class of $d$ intervals in one dimension where the positive examples form a union of intervals. That is $\mathcal{F} = \left\{ \sum_{i=1}^{k} \mathbb{1}_{[a_i, b_i]} : a_1 \leq b_1 \leq \cdots \leq a_k \leq b_k \in [0, 1] \right\}$ over the domain $\mathcal{X} = [0, 1]$. This class has VC dimension $2d$ but infinite star number. Suppose that $d = 2$ but examples in the second interval are very rarely selected by i.i.d. examples. Then the disagreement-based learner would suggest to abstain on all examples to protect against the possibility that our new example is in the second interval. However, consider the following simple strategy: if the new example lies between two positives (resp. negatives), we predict positive (resp. negative), else we abstain.

**Proposition 3.2.** *The proposed strategy for the class of $d$-intervals in one dimension has*

$$\mathsf{MisclassiciationError} \leq d \text{ and } \mathsf{AbstentionError} \leq 2d \log T.$$

To see this, note that whenever we predict, either we are correct or we have identified the location of a new interval, hence reducing the VC dimension of the version space by 1. Since there are at most $d$ intervals, we will therefore make at most $2d$ errors when we predict, implying $\mathsf{MisclassiciationError} \leq 2d$. As for abstaining on i.i.d. examples, the same argument for thresholds can be applied here by treating the intervals as at most $d$ thresholds.

# 4 Higher-order Disagreement-based learner with Known marginals

We will first focus on the setting when the marginal distribution of the i.i.d. distribution $\mathfrak{D}$ is known to the learner. Suppose $\mathcal{F}$ has VC dimension $d$. In this setting, the algorithm that naturally suggests itself is to take a cover for the class under $\mathfrak{D}$ of accuracy $\mathrm{poly}(T^{-1})$ and use an adversarial algorithm for prediction. Since there are covers of size $T^{O(d)}$ and Littlestone dimension of any finite class is bounded by the logarithm of the size, this seems to indicate that this algorithm will achieve our goal with misclassication error $O(d \log T)$ and zero abstention error. But unfortunately note that this algorithm competes only with the best classifier on the cover. The cover is a good approximation only on the marginal distribution $\mathfrak{D}$ and not on the adversarial examples. In fact, when we restrict to the hypothesis class being the cover, the data may no longer even be realizable by the cover. Therefore, we need to use the access to the distribution is a completely different way.

The inspiration for our approach comes from the work of [Han09, Han12] on active learning strategies that go beyond disagreement-based learning by making use of higher-order notions of disagreement based on *shattering*. We note, however, that while the work of [Han09, Han12] only yields asymptotic and distribution-dependent guarantees (and necessarily so, in light of the minimax characterization of [HY15] based on the star number), our analysis differs significantly in order to yield distribution-free finite-sample bounds on the misclassification error and abstention rate.

As we saw earlier, just looking at the disagreement region does not lead to a good algorithm for general VC classes (whenever the star number is large compared to the VC dimension). The main algorithmic insight of this section is that certain higher-order versions of disagreement sets do indeed lead to good notions of uncertainty for VC classes. Our measure uses the probability of shattering $k$ examples (see definition below) for different values of $k$ freshly drawn from the underlying distribution, under the two restrictions of the class corresponding to the two labels for the current test example, to make the abstention decision. One can think of the probability of shattering as a proxy for the complexity of the version space. This serves both as a method to quantify our uncertainty about whether the current example is from the distribution or not, since we can understand the behavior of this quantity in the i.i.d. case, and also as potential function which keeps track of the number of mistakes. In order to formally state this, we will need some definitions.

**Definition 4.1** (Shattering and VC Dimension). Let $\mathcal{X}$ be a domain and $\mathcal{F}$ be a binary function class on $\mathcal{X}$ i.e. $\mathcal{F} \subset \{0, 1\}^{\mathcal{X}}$. A set $\{x_1, \ldots, x_k\} \subseteq \mathcal{X}$ is said to be shattered by $\mathcal{F}$ if for all $y \in \{0, 1\}^k$ there exists a function $f \in \mathcal{F}$ such that $f(x_i) = y_i$. The VC dimension of $\mathcal{F}$ is defined as the maximum $k$ such that there is a set of size $k$ that is shattered.

**Definition 4.2** (Shattered $k$-tuples). Let $k$ be a positive integer. The set of shattered $k$-tuples, denoted by $\mathcal{S}_k$, for hypothesis class $\mathcal{F}$ over a domain $\mathcal{X}$ is defined as

$$\mathcal{S}_k(\mathcal{F}) = \left\{ (x_1, \ldots, x_k) : \{x_1, \ldots x_k\} \text{ is shattered by } \mathcal{F} \right\}.$$

Additionally, given a distribution $\mathfrak{D}$ on the domain, we will refer to as the $k$ shattering probability of $\mathcal{F}$ with respect to $\mathfrak{D}$, denoted by $\rho_k(\mathcal{F}, \mathfrak{D})$, as

$$\rho_k(\mathcal{F}, \mathfrak{D}) = \mathfrak{D}^{\otimes k}\left(\mathcal{S}_k(\mathcal{F})\right) = \Pr_{x_1, \ldots, x_k \sim \mathfrak{D}^{\otimes k}}\left[\{x_1, \ldots, x_k\} \text{ is shattered by } \mathcal{F}\right].$$

Let us now describe the algorithm (see Algorithm 1). The algorithm maintains a state variable $k$ which we will refer to as the level the algorithm is currently in. The level can be thought of as the complexity of the current version space. At level $k$, we will work with shattered sets of size $k$. At each round, the algorithm, upon receiving the example $\hat{x}_t$, computes the probabilities of shattering $k$ examples (drawn i.i.d. from $\mathfrak{D}$) for each of the classes corresponding to sending $\hat{x}_t$ to 0 and 1 respectively. The algorithm abstains if both these probabilities are large, else predicts according to whichever one is larger. At the end of the round, after receiving the true label $y_t$, the algorithm checks whether the probability of shattering $k$ examples is below a threshold $\alpha_k$, in which case it changes the level, that is, updates $k$ to be $k-1$.

---

**Algorithm 1:** Level-based learning for Prediction with Abstension

---
Set $k = d$ and $\mathcal{F}_1 = \mathcal{F}$
**for** $t = 1, \ldots, T \wedge k > 1$ **do**
    Receive $\hat{x}_t$
    **if** $\min\left\{\rho_k\left(\mathcal{F}_t^{\hat{x}_t \to 1}\right), \rho_k\left(\mathcal{F}_t^{\hat{x}_t \to 0}\right)\right\} \geq 0.6\rho_k(\mathcal{F}_t)$ **then** predict $\hat{y}_t = \perp$

    **else** predict $\hat{y}_t = \operatorname{argmax}_{j \in \{0,1\}}\left\{\rho_k\left(\mathcal{F}_t^{\hat{x}_t \to j}\right)\right\}$

    Upon receiving label $y_t$, update $\mathcal{F}_{t+1} \leftarrow \mathcal{F}_t^{\hat{x}_t \to y_t}$
    **if** $\rho_k(\mathcal{F}_{t+1}) \leq \alpha_k$ **then** Set $k = k-1$
**if** $k = 1 \wedge \hat{x}_t \in \mathcal{S}_1$ **then** $\hat{y}_t = \perp$
**if** $k = 1 \wedge \hat{x}_t \notin \mathcal{S}_1$ **then** Predict with the consistent label for $\hat{x}_t$

---

Below, we state the main error bound of the algorithm and give an overview of the analysis. We defer the detailed proofs to Appendix B. Our theorem shows that both the misclassication error and the abstension error are bounded in terms of the VC dimension.

**Theorem 4.1.** *In the adversarial injection model with abstentions with time horizon T, Algorithm 1 with $\alpha_k = T^{-k}$ gives the following,*

$$\mathbb{E}[\mathsf{MisclassificationError}] \leq d^2 \log T \quad and \quad \mathbb{E}[\mathsf{AbstentionError}] \leq 6d.$$

We will now present the main technical idea that we will use to analyze the theorem. Let $k$ be a positive integer and $x$ be any example in $\mathcal{X}$. The following lemma upper bounds the probability for a random example the shattering probability of the two restrictions of the class are both large compared to the original shattering probability of the class. That is to say for most examples, one of the two restrictions of the class will have a smaller shattering probability compared to the original class.

**Lemma 4.2.** *For any $k \in \mathbb{N}$ and any $\eta > 1/2$, we have*

$$\Pr_{x \sim \mathfrak{D}}\left[\rho_k\left(\mathcal{F}^{x \to 1}\right) + \rho_k\left(\mathcal{F}^{x \to 0}\right) \geq 2\eta\rho_k(\mathcal{F})\right] \leq \frac{1}{2\eta - 1} \cdot \frac{\rho_{k+1}(\mathcal{F})}{\rho_k(\mathcal{F})}. \tag{1}$$

With this in hand, we will first look at the abstention error. The intuition is that when the algorithm is at level $k$, an abstention occurs only if the condition from (1) is satisfied and Lemma 4.2 bounds the probability of this event. It remains to note that when the algorithm is at level $k$ we both have an upper bound on $\rho_{k+1}$ (since this is the condition to move down to level $k$ from level $k+1$) and a lower bound on the $\rho_k$ (since this is the condition to stay at level $k$).

**Lemma 4.3** (Abstention error). *For any $k \leq d$, let $[\ell_k, e_k]$ denote the interval of time when Algorithm 1 is at level $k$. Then, the expected number of non-adversarial rounds at level $k$ on which the algorithm abstains satisfies*

$$\mathbb{E}\left[\sum_{t=\ell_k}^{e_k} \mathbb{I}\left[c_t = 0 \wedge \hat{y}_t = \perp\right]\right] \leq 5T \cdot \frac{\alpha_{k+1}}{\alpha_k}. \tag{2}$$

Next, we bound the misclassification error. The main idea here is to note that every time a misclassification occurs at level $k$, the $k$-th disagreement coefficient reduces by a constant factor. Since we have a lower bound on the disagreement coefficient at a fixed level, this leads to a logarithmic bound on the number of misclassifications at any given level.

**Lemma 4.4** (Misclassification error). *For any $k \leq d$, let $[\ell_k, e_k]$ denote the interval of time when Algorithm 1 is at level $k$. For any threshold $\alpha_k$ in Algorithm 1,*

$$\mathbb{E}\left[ \sum_{t=\ell_k}^{e_k} \mathbb{I}\left[ \hat{y}_t = 1 - f^*(\hat{x}_t) \right] \right] \leq 2 \cdot \log\left( \frac{1}{\alpha_k} \right).$$

Putting together Lemma 4.4 and Lemma 4.3 along with a setting of $\alpha_k = T^{-k}$, gives us Theorem 4.1.

*Remark.* Closer inspection at the algorithm and the analysis reveals that the algorithm gives the same guarantees even in the setting where the learner does not receive the label $y_t$ during rounds that it abstains (hinted at in Section 2). To see this note that during the rounds that we abstain, we only use the fact that $\rho_k\left(\mathcal{F}_t\right)$ does not increase which would remain true when if we did not update the class.

# 5 Structure-based Algorithm for Unknown Distribution

We move to the case of unknown distributions. In this setting, the example $x$ are drawn from a distribution $\mathfrak{D}$ that is unknown to the learner. As we saw earlier, it is challenging to decide whether a single point is out of distribution or not, even when the distribution is known. This current setting is significantly more challenging since the learner needs to abstain on examples that are out of distribution for a distribution that it doesn't know. A natural idea would be to use the historical data to build a model for the distribution. The main difficulty is that, since we do not get feedback about what examples are corrupted, our historical data has both in-distribution and out-of-distribution examples. The only information we have about what examples are out of distribution is the prediction our algorithm has made on them. Such issues are a major barrier in moving from the known distribution case to the unknown distribution case. In this section, we will design algorithms for two commonly studied hypothesis classes: VC dimension 1 classes, and axis-aligned rectangles. Though, VC dimension 1 seems restrictive, there are classes with VC dimension 1 with large Littlestone dimension and large star number, which capture the complexity of sequential prediction and active learning respectively. Thus, our results separate the complexity of those models from our model of abstention.

## 5.1 Structure-based Algorithm for VC Dimension 1 Classes

A key quantity that we will use in our algorithm will be a version of the probability of the disagreement region but computed on the historical data. The most natural version of this would be the leave-one-out disagreement. That is, consider the set of examples which are in the disagreement region when the class is restricted using the data set with the point under consideration removed. This estimate would have been an unbiased estimator for the disagreement probability (referred to earlier as $\rho_1$ in Definition 4.2). As mentioned earlier, unfortunately, in the presence of adversarial injections, this need not be a good estimate for the disagreement probability.

In order to remedy this, we consider a modified version of the leave-one-out disagreement estimate which considers examples $x$ in the disagreement region for the class $\mathcal{F}|_{S_f \setminus (x,y)}$ where $S_f$ is the subset of the datapoints which disagrees with a fixed reference function $f$. It is important to note that this function $f$ is fixed independent of the true labelling function $f^\star$. Though this seems a bit artificial at first, but this turns out to be a natural quantity to consider given the structure theorem for VC dimension one classes (see Appendix C.1 for more details).

**Definition 5.1.** Let $\mathcal{F}$ be a class of functions and let $f \in \mathcal{F}$ be a reference function. Let $S = \left\{ (x_i, y_i) \right\}$ be a realizable data set. Define

$$\Gamma(S, \mathcal{F}, f) = \left\{ x : \exists y \quad (x, y) \in S \wedge x \in \mathcal{S}_1\left( \mathcal{F}|_{S_f \setminus (x,y)} \right) \right\}$$

where $S_f = \{ (x, y) \in S : f(x) \neq y \}$. Further, we will denote the size of this set by $\gamma(S, \mathcal{F}) = \left| \Gamma(S, \mathcal{F}, f) \right|$. We will suppress the dependence on $\mathcal{F}$ when it is clear from context.

Let us now present the main algorithm (see Algorithm 2). The main idea of the algorithm is similar to Algorithm 1 in the known distribution case. We will make a prediction in the case when the difference between $\Gamma$ for the two classes corresponding to the two labels for the point $x$ is large. The idea is that when a prediction is made and a mistake happens, the size of $\Gamma$ goes down similar to $\rho_k$ in the known distribution case. In the case when the difference is small, we will abstain.

---

**Algorithm 2:** Structure-based learning for Prediction with Abstention for Unknown Distribution

---

Let $f \in \mathcal{F}$ be a reference function and $\alpha$ be the abstention threshold.
Set $\mathcal{F}_0 = \mathcal{F}$ and $S_0 = \emptyset$
**for** $t = 1, \ldots, T$ **do**
    Receive $\hat{x}_t$
    Let $a_0 = \left| \Gamma \left( S_{t-1}, \mathcal{F}^{\hat{x}_t \to 0}, f \right) \right|$ and $a_1 = \left| \Gamma \left( S_{t-1}, \mathcal{F}^{\hat{x}_t \to 1}, f \right) \right|$
    **if** $\hat{x}_t \notin \mathcal{S}_1 \left( \mathcal{F}_t \right)$ **then** Predict with the consistent label for $\hat{x}_t$
    **else if** $\max \{a_0, a_1\} \geq \alpha$ **then** $\hat{y}_t = \operatorname{argmax}_b a_b$
    **else** $\hat{y}_t = \bot$
    Upon receiving label $y_t$, update $S_t \leftarrow S_{t-1} \cup \left\{ (\hat{x}_t, y_t) \right\}$ and $\mathcal{F}_t \leftarrow \mathcal{F}_{t-1}^{\hat{x}_t \to y_t}$

---

Though the algorithm is simple and can be made fairly general, our analysis is restricted to the case of VC dimension one. The main reason for this restriction is that our analysis relies on a structure theorem for VC dimension one classes which has no direct analogy for higher VC dimension. But since the algorithm has a natural interpretation independent of this representation, we expect similar algorithms to work for higher VC dimension classes as well.

**Theorem 5.1.** *Let $\mathcal{F}$ be a hypothesis class with VC dimension $1$. Then, in the corruption model with abstentions with time horizon T, Algorithm 2 with parameter $\alpha = \sqrt{T}$ gives the following guarantee*

$$\mathbb{E}[\mathsf{MisclassificationError}] \leq 2\sqrt{T \log T} \quad and \quad \mathbb{E}[\mathsf{AbstentionError}] \leq \sqrt{T \log T}.$$

We will now present the main technical ideas that we use to analyze the algorithm. We will keep track of the mistakes using the size of the disagreement region, which we denote by $\gamma_t = \gamma \left( S_t, \mathcal{F} \right)$. The main idea for the proof is to note that when we decide to predict the label with the larger value of $\Gamma$ is bigger by an additive $\alpha$. Thus, when a mistake occurs the value of $\gamma$ decreases by at least $\alpha$. Summing the errors over all time steps gives the following bound.

**Lemma 5.2.** *Algorithm 2 has $\mathbb{E}[\mathsf{MisclassificationError}] \leq 2T/\alpha$.*

Next, we move on to the number of abstentions. In fact, we will prove a stronger structural results that shows that the number of examples such that there is any set of adversarial injections that would make the algorithm abstain is small. We refer to examples on which algorithm can be made to abstain as attackable examples (formally defined in Definition C.2). The main idea is to prove that in any set of iid examples, there are only a few attackable ones. This is formally stated and proved as Lemma C.3. Using this claim, we can bound the number of abstentions using an exchangeability argument.

**Lemma 5.3.** *Algorithm 2 has $\mathbb{E}[\mathsf{AbstentionError}] \leq \alpha \log T$.*

The proof of Theorem 5.1 follows from Lemma 5.2 and Lemma 5.3 by setting $\alpha = \sqrt{T/\log T}$.

## 5.2 Structure-based Algorithm for Axis-aligned Rectangles

We can extend the idea from above to design an algorithm for the class of axis-aligned rectangles in dimension $p$ (VC dimension is $2p$) that achieves abstention and misclassification error bounded by $O(p\sqrt{T \log T})$. This exhibits a class of VC dimension $> 1$ for which we can attain the desired guarantees without access to the distribution.

**Theorem 5.4.** *Let $\mathcal{F}$ be the class of axis aligned rectangles in $\mathbb{R}^p$. Then, Algorithm 3 with $\alpha = \sqrt{T/\log T}$ satisfies*

$$\mathsf{MisclassificationError} \leq p\sqrt{T \log T},$$
$$\mathsf{AbstentionError} \leq 2p\sqrt{T \log T} + 2p \log T.$$

For details of the algorithm and analysis, we refer the reader to Appendix C.3.1.

# 6 Discussion and Future Directions

In this paper, we introduce a framework for beyond-worst case adversarial sequential predictions by using abstention. Our proposed framework falls at the intersection of several learning paradigms such as active learning, uncertainty quantification, and distribution testing. In this framework we show two main positive results, validating the beyond-worse case nature of our framework. However, our work has only scratched the surface of understanding learnability for VC classes in this new framework. Here we discuss several exciting future directions:

**General VC classes with unknown distribution.** Extending the structure-based algorithms for general VC classes is wide open. Even characterizing the complexity of learning in the unknown distribution is open. One could attempt to convert the algorithm from Section 4 that for any class $\mathcal{F}$ with VC dimension $d$, achieves abstention error and misclassification error bounded only as a function of $d$. Recall that Algorithm 1 computed the probabilities of shattering $k$ points using the knowledge of the distribution and made a prediction $\hat{y}_t$ depending on the relative magnitudes of the probabilities corresponding to the two restricted classes. The main challenge in the unknown distribution case is that it is not immediately obvious how to compute these quantities.

One natural approach is to use the historical data as a proxy for the distribution. That is, given the data set $S_t$ of size $n$, compute the leave-$k$-out estimator for the probability as follows $\tilde{\rho}_k(S, \mathcal{F}) = 1/\binom{n}{k} \sum_{T \subset S; |T|=k} \mathbb{1}\left[T \text{ is shattered by } \mathcal{F}|_{S \setminus T}\right]$. There are a few things to observe about this estimator. First, in the case when the data is generated i.i.d., this estimator is unbiased. Further, though each of the summands is not independent, one can show concentration for estimators of this form using the theory of U-statistics. Additionally, recall that in Algorithm 1 required the thresholds $\alpha_k$ to be set to $T^{-O(k)}$ (we use $T^{-k}$ but it is straightforward to extend this to $T^{-ck}$ for $c < 1$). This appears to be high precision but note that the "number of samples" one has for a data set of size $n$ is $n^{O(k)}$. Thus, it is conceivable that such an estimator can give the necessary bounds. Unfortunately, the challenge with analyzing this in our setting is that we do no know which of the examples are adversarial. Thus, the adversary could inject examples to make our estimates arbitrarily inaccurate. Thus, as in the case of the VC dimension 1 classes and rectangles, our analysis would need to not rely on the accuracy of the estimates but rather use these estimates to maintain progress or construct other versions of estimators that are unaffected by the adversarial injections.

**Tight bounds for known distribution.** In the known distribution case, it remains to find the optimal error bound. It would be interesting to improve the upper bound or perhaps even more interesting to show a separation between stochastic sequential prediction and our model. We conjecture that the correct dependence on VC dimemsion $d$ should be linear and not quadratic, and our analysis is potentially lose in this aspect. A potential strategy to obtain an improved bound would be to choose the level $k$ at each iteration in an adaptive manner.

**Beyond realizability.** Our techniques rely strongly on the realizability (no label noise), however our framework can naturally be extended to settings with label noise. Immediate questions here would be to extend our results to simple noise models such as random classification noise and Massart noise or more ambitiously the agnostic noise model.

**Beyond binary classification.** Our framework can naturally be extended to more general forms of prediction such as multiclass classification, partial concept classes, and regression. It would be interesting to characterize the complexity of learning in these settings.

**Connections to conformal prediction and uncertainty quantification.** The unknown distribution case can be seen as form of distribution-free uncertainty quantification. It would be interesting to understand connections to other forms such as conformal prediction. On a technical level, our work exploits exchangeability of the i.i.d. sequence which is the foundation of conformal prediction, though the main challenge in our setting is the presence of adversarial inputs. It would be interesting to build on this connections and understand whether techniques can be ported over in either direction.

**Computationally efficient algorithms.** Our focus for this paper has been entirely on the statistical benefits of abstention. Understanding the computational complexity in this setting is an exciting avenue for research. Concretely, for halfspaces in $d$ dimensions, is there a polynomial time algorithm for learning with abstentions, even for well-behaved distributions such as Gaussians. On a related note, showing computational-statistical gaps in this specialized setting would be interesting, albeit disappointing.

## Acknowledgements

S. Goel would like to sincerely thank Thodoris Lykouris and Adam Tauman Kalai for working with her on designing the model of adversarial learning with abstention. Several ideas presented in the Section 2 and 3 are based on discussions with T. Lykouris and A. Kalai. We would also like to thank our anonymous NeurIPS reviewer for helping us improve the clarity and accessibility of our paper.

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

# A Preliminaries

We introduce the framework of sequential binary prediction. We focus on the realizable case for direct comparison to the model that we will subsequently introduce. In sequential binary prediction, at time step $t$ an adversary picks a $x_t \in \mathcal{X}$ (potentially from a distribution $\mathfrak{D}_t$) and presents it to the learner. The learner then picks $\hat{y}_t \in \{0, 1\}$ and then receives as label $y_t = f^\star(x_t)$ where $f \in \mathcal{F}$ is the unknown function that the learner is trying to learn. The objective of the learner is to minimize the number of mistakes it makes. This is measured using the regret (also known as the mistake bound in the realizable case), defined as

$$\mathrm{Regret}_T = \sum_{t=1}^{T} \mathbb{1}\left[\hat{y}_t \neq y_t\right].$$

The stochastic case of sequential binary prediction corresponds to the setting when the input $x_t$ is sampled from a fixed $\mathfrak{D}_t = \mathfrak{D}$ independently across time. This is the sequential analogue of the classical i.i.d. batch model of learning. We now introduce a fundamental notion of complexity for hypothesis classes, the VC dimension which captures the regret in this setting. The VC dimension measures how expressive a hypothesis class is by measuring the number of functions that can be realized by the class on a set of points

**Definition A.1** (Shattering and VC Dimension). Let $\mathcal{X}$ be a domain and $\mathcal{F}$ be a binary function class on $\mathcal{X}$ i.e. $\mathcal{F} \subset \{0, 1\}^{\mathcal{X}}$. A set $\{x_1, \ldots, x_k\} \subseteq \mathcal{X}$ is said to be shattered by $\mathcal{F}$ if for all $y \in \{0, 1\}^k$ there exists a function $f \in \mathcal{F}$ such that $f(x_i) = y_i$. The VC dimension of $\mathcal{F}$ is defined as the maximum $k$ such that there is a set of size $k$ that is shattered and is denoted by $\mathrm{VCDim}(\mathcal{F})$.

The VC dimension was originally introduced in the context of binary classification in the batch setting but minor modifications of the argument show that even in the sequential prediction setting, the VC dimension characterizes the regret in stochastic sequential prediction.

Continuing the discussion about shattering, we will introduce a notion that corresponding to the probability of shattering a set of points. Though this notion does feature in usual analysis of sequential prediction, it will be a key to the design and analysis of algorithms in our setting. See Section 4 for further discussion.

**Definition A.2** (Shattered $k$-tuples). Let $k$ be a positive integer. The set of shattered $k$-tuples, denoted by $\mathcal{S}_k$, for hypothesis class $\mathcal{F}$ over a domain $\mathcal{X}$ is defined as

$$\mathcal{S}_k(\mathcal{F}) = \left\{(x_1, \ldots, x_k) : \{x_1, \ldots x_k\} \text{ is shattered by } \mathcal{F}\right\}.$$

Additionally, given a distribution $\mathfrak{D}$ on the domain, we will refer to as the $k$ shattering probability of $\mathcal{F}$ with respect to $\mathfrak{D}$, denoted by $\rho_k(\mathcal{F}, \mathfrak{D})$, as

$$\rho_k(\mathcal{F}, \mathfrak{D}) = \mathfrak{D}^{\otimes k}\left(\mathcal{S}_k(\mathcal{F})\right) = \Pr_{x_1, \ldots, x_k \sim \mathfrak{D}^{\otimes k}}\left[\{x_1, \ldots, x_k\} \text{ is shattered by } \mathcal{F}\right].$$

A case of particular interest in the above definition is $k = 1$. This notion is closely studied in active learning under the name of disagreement region.

**Definition A.3** (Disagreement Region). Let $\mathcal{F}$ be a hypothesis class on domain $\mathcal{X}$. The disagreement region of $\mathcal{F}$ is defined as

$$\mathcal{S}_1(\mathcal{F}) = \left\{x \in \mathcal{X} : \exists f, g \in \mathcal{F} \text{ such that } f(x) \neq g(x)\right\}.$$

Analogously, we will the disagreement probability as

$$\rho_1(\mathcal{F}, \mathfrak{D}) = \mathfrak{D}\left(\mathcal{S}_1(\mathcal{F})\right) = \Pr_{x \sim \mathfrak{D}}\left[\exists f, g \in \mathcal{F} \text{ such that } f(x) \neq g(x)\right].$$

Going back to sequential prediction, in a seminal result [Lit87] showed that the mistake bound is characterized by a combinatorial property of the hypothesis class known as the Littlestone dimension. Though the definition would not be central to our work, we include it here for comparison to our results.

**Definition A.4** (Littlestone Dimension). Let $\mathcal{F}$ be a hypothesis class on domain $\mathcal{X}$. A mistake tree is a full binary decision of depth $\ell$ tree whose internal nodes are labelled by elements of $\mathcal{X}$. Every root

to leaf path in the mistake tree corresponds to a sequence $(x_i, y_i)_{i=1}^{\ell}$ by associating a label $y_i$ to a node depending on whether it is the left or right child of its parent on the path. A mistake tree of depth $\ell$ is said to be shattered by a class $\mathcal{F}$ if for any root to leaf path $(x_i, y_i)_{i=1}^{\ell}$, there is a function $f \in \mathcal{F}$ such that $f(x_i) = y_i$ for all $i \leq \ell$. The Littlestone dimension of the class $\mathcal{F}$ denoted by LDim $(\mathcal{F})$ is the largest depth of a mistake tree shattered by the class $\mathcal{F}$.

The Littlestone dimension completely characterizes the mistake bound of the class $\mathcal{F}$ in the sense that any algorithm can be forced to have a mistake a bound of $\Omega \left( \text{LDim} \left( \mathcal{F} \right) \right)$ and there exists an algorithm known as the standard optimal algorithm (SOA) with a mistake bound of $O \left( \text{LDim} \left( \mathcal{F} \right) \right)$. Though this characterization is elegant, it is usually interpreted as a negative result since even simple classes the Littlestone dimension is infinite. This is exemplified by the class of thresholds i.e. $\mathcal{F} = \left\{ \mathbb{1}_{[a,1]} : a \in [0, 1] \right\}$ over the domain $\mathcal{X} = [0, 1]$, for which the VC dimension is 1 but the Littlestone dimension is infinite. As noted earlier, the VC dimension captures the number of mistakes when the inputs are i.i.d. while Littlestone captures the number of mistakes for the worst case, dealing to a disparity between achievable mistake bounds in the two settings. This discrepancy is the main motivation for the study of sequential prediction in beyond worst-case settings.

# B Proofs from Section 4

## B.1 Properties of Higher-Order Disagreement: Proof of Lemma 4.2

We first begin by relating the probabilities of shattering $k$ points for the classes gotten by restricting to evaluating to 0 and 1 at $x$ respectively, for an arbitrary point $x$. The proof of the following lemma uses a simple inclusion exclusion argument.

**Lemma B.1.** *For all $x \in \mathcal{X}$ and any hypothesis class $\mathcal{F}$, we have*

$$\mathfrak{D}^{\otimes k} \left( \mathcal{S}_k \left( \mathcal{F}^{x \to 1} \right) \right) + \mathfrak{D}^{\otimes k} \left( \mathcal{S}_k \left( \mathcal{F}^{x \to 0} \right) \right) \leq \mathfrak{D}^{\otimes k} \left( \mathcal{S}_k \left( \mathcal{F} \right) \right) + \mathfrak{D}^{\otimes k} \left( \mathcal{S}_k \left( \mathcal{F}^{x \to 1} \right) \cap \mathcal{S}_k \left( \mathcal{F}^{x \to 0} \right) \right).$$

*Equivalently,*

$$\rho_k \left( \mathcal{F}^{x \to 1} \right) + \rho_k \left( \mathcal{F}^{x \to 0} \right) \leq \rho_k \left( \mathcal{F} \right) + \Pr_{x_1, \ldots, x_k \sim \mathfrak{D}} \left[ x_1, \ldots, x_k \text{ is shattered by both } \mathcal{F}^{x \to 1} \text{ and } \mathcal{F}^{x \to 0} \right].$$

*Proof.* For $x_1, \ldots, x_n \in \mathcal{X}$, consider the four indicator random variables given by

$$A_1 = \mathbb{1} \left( x_1, \ldots, x_n \text{ is shattered by } \mathcal{F}^{x \to 1} \right)$$

$$A_2 = \mathbb{1} \left( x_1, \ldots, x_n \text{ is shattered by } \mathcal{F}^{x \to 0} \right)$$

$$A_3 = \mathbb{1} \left( x_1, \ldots, x_n \text{ is shattered by } \mathcal{F}^{x \to 1} \text{ and } \mathcal{F}^{x \to 0} \right)$$

$$A_4 = \mathbb{1} \left( x_1, \ldots, x_n \text{ is shattered by } \mathcal{F} \right).$$

Note that $A_1 + A_2 \leq A_3 + A_4$. Taking expectations gives the desired result. $\qquad \square$

In order to prove the main lemma, we then take expectations with respect to the point $x$ drawn independently from $\mathfrak{D}$. The key observation is to relate the probability of $k$ point being shattered by both classes that evaluate to 0 and 1 at $x$, for a random point $x$, to the probability of shattering $k + 1$ points.

*Proof of Lemma 4.2.* The proof follows by using Lemma B.1 to get (3) and Markov's inequality to get (4). The final line follows by noting that since $x_1, \ldots, x_k$ and $x$ are drawn from $\mathfrak{D}$ and are

shattered, (4) computes the probability that $k + 1$ points are shattered.

$$\Pr_{x \sim \mathfrak{D}} \left[ \rho_k \left( \mathcal{F}^{x \to 0} \right) + \rho_k \left( \mathcal{F}^{x \to 1} \right) \geq 2\eta \rho_k \left( \mathcal{F} \right) \right]$$

$$\leq \Pr_{x \sim \mathfrak{D}} \left[ \rho_k \left( \mathcal{F} \right) + \Pr_{x_1, \ldots, x_k \sim \mathfrak{D}} \left[ x_1, \ldots, x_k \text{ is shattered by both } \mathcal{F}^{x \to 1} \text{ and } \mathcal{F}^{x \to 0} \right] \geq 2\eta \rho_k \left( \mathcal{F} \right) \right] (3)$$

$$\leq \Pr_{x \sim \mathfrak{D}} \left[ \Pr_{x_1, \ldots, x_k \sim \mathfrak{D}} \left[ x_1, \ldots, x_k \text{ is shattered by both } \mathcal{F}^{x \to 1} \text{ and } \mathcal{F}^{x \to 0} \right] \geq (2\eta - 1) \rho_k \left( \mathcal{F} \right) \right]$$

$$\leq \frac{\cdot \mathbb{E} \left[ \Pr_{x_1, \ldots, x_k \sim \mathfrak{D}} \left[ x_1, \ldots, x_k \text{ is shattered by both } \mathcal{F}^{x \to 1} \text{ and } \mathcal{F}^{x \to 0} \right] \right]}{(2\eta - 1) \rho_k \left( \mathcal{F} \right)} \tag{4}$$

$$\leq \frac{1}{2\eta - 1} \cdot \frac{\rho_{k+1} \left( \mathcal{F} \right)}{\rho_k \left( \mathcal{F} \right)}.$$

$\square$

### B.2   Proof of Lemma 4.3

*Proof.* Let $t \in [\ell_k, m_k]$. Recall that $\mathcal{F}_t$ denotes the class consistent with the data seen till time $t$. Recall that $H_t$ denotes the history of the interaction till time $t$.

$$\mathbb{E} \left[ \mathbb{1} \left[ c_t = 0 \wedge \hat{y}_t \left( \hat{x}_t \right) = \perp \right] \mid H_t \right] = \mathbb{E} \left[ \mathbb{1} \left[ c_t = 0 \wedge \hat{y}_t \left( x_t \right) = \perp \right] \mid H_t \right] \tag{5}$$

$$\leq \mathbb{E} \left[ \mathbb{1} \left[ \hat{y}_t \left( x_t \right) = \perp \right] \mid H_t \right]$$

$$\leq \Pr \left[ \min \left\{ \rho_k \left( \mathcal{F}_t^{x_t \to 1} \right), \rho_k \left( \mathcal{F}_t^{x_t \to 1} \right) \right\} \geq 0.6 \rho_k \left( \mathcal{F}_t \right) \mid H_t \right] \tag{6}$$

$$\leq \Pr \left[ \rho_k \left( \mathcal{F}_t^{x_t \to 1} \right) + \rho_k \left( \mathcal{F}_t^{x_t \to 1} \right) \geq 1.2 \rho_k \left( \mathcal{F}_t \right) \mid H_t \right] \tag{7}$$

$$\leq 5 \frac{\rho_{k+1} \left( \mathcal{F}_t \right)}{\rho_k \left( \mathcal{F}_t \right)} \tag{8}$$

$$\leq 5 \frac{\alpha_{k+1}}{\alpha_k}. \tag{9}$$

The equality in (5) follows from the fact that in the uncorrupted rounds, $\hat{x}_t = x_t$. The inequality in (6) follows from the condition for abstention in Algorithm 1 and the fact that $x_t \sim \mathfrak{D}$. (7) follows from the fact that the min of two numbers is at most their average. The key step is (8) which follows from Lemma 4.2. (9) follows from the fact that at level $k$ in Algorithm 1, $\rho_{k+1} \left( \mathcal{F}_t \right) \leq \alpha_k$ and $\rho_k \left( \mathcal{F}_t \right) \geq \alpha_{k+1}$. Summing this bound and noting that $e_k - \ell_k$ is at most $T$, we get the required bound. $\square$

### B.3   Proof of Lemma 4.4

*Proof.* Note that from definition of Algorithm 1, we have that when $\hat{y}_t \neq \perp$, $\min \left\{ \rho_k \left( \mathcal{F}_t^{x_t \to 1} \right), \rho_k \left( \mathcal{F}_t^{x_t \to 0} \right) \right\} \leq 0.6 \rho_k \left( \mathcal{F}_t \right)$. Thus, since we predict with the label corresponding to $\max \left\{ \rho_k \left( \mathcal{F}_t^{x_t \to 1} \right), \rho_k \left( \mathcal{F}_t^{x_t \to 0} \right) \right\}$, if we make a mistake, we have

$$\rho_k \left( \mathcal{F}_{t+1} \right) \leq 0.6 \cdot \rho_k \left( \mathcal{F}_t \right).$$

Also, note once end of the phase corresponding to $k$, $e_k$, is reached when $\rho_k \left( \mathcal{F}_t \right) \leq \alpha_k$. This leads to the required bound. $\square$

### B.4   Proof of Theorem 4.1

*Proof.* First, let us look at the misclassification error. Note that Algorithm 1 will not misclassify when $k = 1$. To see this, recall from Definition 4.2 that $\hat{x}_t \not{S}_1$ implies that there is a unique label

consistent with the history. For the remaining levels, we sum the errors from Lemma 4.4 and recall that $\alpha_k = T^{-k}$, which gives us

$$\mathsf{MisclassificationError} = \sum_{t=1}^{T} \mathbb{1}[\hat{y}_t = 1 - f^{\star}(x_t)]$$

$$\leq 2 \sum_{k=2}^{d} \log\left(\frac{1}{\alpha_k}\right)$$

$$\leq 2 \sum_{k=2}^{d} k \log T$$

$$\leq d^2 \log T.$$

For the abstention error, we again begin with the case of $k = 1$. Note that for $t \geq \ell_1$, we have

$$\Pr_{x \sim \mathfrak{D}} \left[x \in \mathcal{S}_1\left(\mathcal{F}_t\right)\right] \leq \alpha_1.$$

Thus, we have a bound of $T\alpha_1 \leq 1$ on the expected error in this case. For the remaining levels, we sum the error from (2) over all $k$, which gives us the bound

$$\mathsf{AbstentionError} = \sum_{t=1}^{T} \mathbb{1}[c_t = 0 \wedge \hat{y}_t = \perp]$$

$$= \sum_{k=1}^{d} \mathbb{E}\left[\sum_{t=\ell_k}^{e_k} \mathbb{1}\left[c_t = 0 \wedge \hat{y}_t = \perp\right]\right]$$

$$= 1 + 5T \sum_{k=2}^{d} \frac{\alpha_{k+1}}{\alpha_k}$$

$$= 1 + 5T \sum_{k=2}^{d} \frac{1}{T}$$

$$\leq 5d + 1$$

$$\leq 6d.$$

This gives us the desired bounds. $\qquad\qquad\square$

## C   Proofs from Section 5

### C.1   Structure for VC Dimension One Classes

**Definition C.1.** Consider a domain $\mathcal{X}$ and a partial order $\prec$ on $\mathcal{X}$. We say that a set $I$ is an initial segment of $\prec$ if for all $x \in I$ and $y \in \mathcal{X}$ such that $y \prec x$, we have $y \in I$. We say that a partial order is a tree ordering if every initial segment $I$ is a linear order i.e. for all $x, y \in I$, either $x \prec y$ or $y \prec x$.

**Theorem C.1** ([BD15]). *Let $\mathcal{F}$ be a hypothesis class over the domain $\mathcal{X}$. Then, the following are equivalent:*

  *a. $\mathcal{F}$ has VC dimension $1$.*

  *b. There is a tree ordering $\prec$ on $\mathcal{X}$ and a hypothesis $f \in \mathcal{F}$ such that every element of the set*

$$\mathcal{F}_f = \{h \oplus f : h \in \mathcal{F}\}$$

  *is an initial segment of $\prec$.*

The result above was initially observed in [BD15].

## C.2 Proof of Lemma 5.2

**Lemma C.2.** *For any t, we have that*

$$\gamma_{t+1} \leq \gamma_t - \alpha \cdot \mathbb{1}\left[\textit{Misclassification at time } t\right] + 1. \tag{10}$$

*Proof.* First, note that in any round that a mistake was not made, we have that $\gamma_{t+1} \leq \gamma_t + 1$. This is because at most one point is added to the data set in each round.

Note that $\left|\Gamma\left(S_{t-1}, \mathcal{F}^{\hat{x}_t \to 1}\right)\right| + \left|\Gamma\left(S_{t-1}, \mathcal{F}^{\hat{x}_t \to 0}\right)\right| \leq \left|\Gamma\left(S_{t-1}, \mathcal{F}\right)\right|$. Further, we have $\left|\Gamma\left(S_t, \mathcal{F}\right)\right| \leq \left|\Gamma\left(S_{t-1}, \mathcal{F}^{\hat{x}_t \to y_t}\right)\right| + 1$. From the condition for predicting, we have that

$$\max\left\{\left|\Gamma\left(S_{t-1}, \mathcal{F}^{\hat{x}_t \to 1}\right)\right|, \left|\Gamma\left(S_{t-1}, \mathcal{F}^{\hat{x}_t \to 0}\right)\right|\right\} \geq \alpha.$$

When we make a misclassication, we go to the smaller value for $\Gamma$, therefore we get the desired bound. $\square$

*Proof of Lemma 5.2.* Note that $\mathsf{MisclassificationError} = \sum_{t=1}^{T} \mathbb{1}\left[\text{Misclassification at time } t\right]$. Rearranging (10) and summing gives us

$$\mathsf{MisclassificationError} \leq \frac{1}{\alpha}\sum_{t=1}^{T}\left(\gamma_t - \gamma_{t+1} + 1\right)$$

Note that for all $i$, we have that $\gamma_i \leq T$ which gives us the desired bound. $\square$

## C.3 Proof of Lemma 5.3

**Definition C.2** (Attackable Point)**.** Let $\mathcal{F}$ be a hypothesis class and let $f \in \mathcal{F}$ be a representative function. Let $S$ be a realizable dataset. We say that a point $x$ is attackable with respect to a data set $S$ if there exists is a sequence of adversarial examples $A_x$ such that algorithm abstains on example $x$ when the history is $S \cup A_x \setminus \{x\}$. In other words, $x$ is attackable if there is a set of adversarial examples $A_x$ such that $x \notin \mathcal{S}_1\left(\mathcal{F}|_{S \cup A_x \setminus \{x\}}\right)$ and

$$\max\left\{\left|\Gamma\left(S \cup A_x \setminus \{x\}, \mathcal{F}^{x \to 0}\right)\right|, \left|\Gamma\left(S \cup A_x \setminus \{x\}, \mathcal{F}^{x \to 1}\right)\right|\right\} \leq \alpha.$$

The key lemma for our analysis is that the number of attackable examples is bounded.

**Lemma C.3.** *Let $\mathcal{F}$ be a hypothesis and $f \in \mathcal{F}$ be any reference function. Let $S$ be any set of examples. Then, the number of attackable examples is at most $\alpha$.*

The proof uses a structure theorem for classes with VC dimension one which states that they can be represented as initial segments of a tree order.

*Proof.* Let $f$ be the representative function from the characterization in Theorem C.1. Futher, let $\mathfrak{T}$ be the tree corresponding to the tree order on $\mathcal{X}$. Since $f$ is a fixed function that does not depend on the algorithm or the history of interaction with adversary, we can preprocess all points and labels to be xored with the labels of $f$. In other words, we tranform the class to be such that $f$ is the all zeros function. In this setting, the true hypothesis $f^\star$ corresponds to a path $p$ and a threshold $x^\star$ on $\mathfrak{T}$ such that $f^\star(x) = 1$ if and only if $x \in p$ and $x \prec x^\star$. For this proof, it is important to consider only adversaries that do not get to chose the true hypothesis $f^\star$ adaptively. Thus the path is fixed throughout the history of the interaction.

Let $S$ be the data set under consideration. First note that by definition, we only need to consider the points $x$ in the disagreement region. The labels of all points that are not in the subtree of the deepest 1 labeled point in $S$ are fixed and thus cannot be in the disagreement region. Thus, we only need to consider the points in the subtree of the deepest 1 labeled point in $S$.

First note that points that have more than $\alpha$ points in descendant subtree cannot be attacked. This is because if the point is labelled as a 0 then all its descendants are in the disagreement region. This remains true even for any points that are added to the data set. Thus, $\left|\Gamma\left(S \cup A_x \setminus \{x\}, \mathcal{F}^{x \to 0}\right)\right| > \alpha$. This is because in the definition of $\gamma$ we remove all points labelled 0.

For any node $u$, denote by $\mathrm{pos}(v)$ the closest ancestor on the path $p$ corresponding to the positive points. We claim that if $u$ has fewer than $\alpha$ points in its descendant subtree, then $u$ is attackable only if

$$\left|\left\{v \in S : \mathrm{pos}(u) \preceq \mathrm{pos}(v) \wedge v \text{ is not a descendant of } u\right\}\right| \leq \alpha \tag{11}$$

First note that adding any 0 labelled points to $S$ as $A_u$ does not change the number of points in $\Gamma$. Further note that for $v$ such that $\mathrm{pos}(u) \preceq \mathrm{pos}(v)$, adding a 1 labelled point must be on the path between $\mathrm{pos}(u)$ and $\mathrm{pos}(v)$. But, this would remove $u$ from the disagreement region. Thus, all point $v$ would be counted in $\Gamma$ and if these are greater than $\alpha$ then $u$ is not attackable.

Consider the point $w$ that is minimal amongst $\mathrm{pos}(u)$ for $u$ satisfying (11) and $u$ be a node such that $\mathrm{pos}(u) = w$. First note that all points that satisfy (11) are in the subtree of $w$. Second, note that this subtree has at most $\alpha$ points. This is because if there were more than $\alpha$ points, $u$ would not satisfy (11) which is a contradiction. $\qquad\square$

*Proof of Lemma 5.3.* Let $i_T$ be the number of i.i.d. points in the data set at time $T$. Note that the only i.i.d. points that we abstain on are the attackable points. But, since the i.i.d. points are exchangeable if $i$ i.i.d. points are seen so far, the probability of abstain is given by $\frac{\alpha}{i}$. Thus, the expected total number of abstentions is at most

$$\sum_{i=1}^{i_T} \frac{\alpha}{i} \leq \alpha \log T$$

as required. $\qquad\square$

### C.3.1 Structure-based Algorithm for Axis-aligned rectangles

Recall that the class of axis-aligned rectangles consists of functions $f$ parameterized by $(a_1, b_1, \ldots, a_p, b_p)$ such that

$$f_{(a_1, b_1, \ldots, a_p, b_p)}(x) = \begin{cases} 1 & \text{if } \forall i \in [p], a_i \leq \hat{x}_i \leq b_i \\ 0 & \text{otherwise.} \end{cases}$$

Now consider the following algorithm:

---
**Algorithm 3:** Structure-based learning for Axis-aligned Rectangles

---
Set $a_1, \ldots, a_p = -\infty$ and $b_1, \ldots, b_p = \infty$
**for** $t = 1, \ldots, T$ **do**
    Receive $\hat{x}_t$
    **if** $\forall \tau < t, y_\tau = 0$ **then** $\hat{y}_t = 0$
    **else if** $\hat{x}_t \notin \mathcal{S}_1(\mathcal{F}_{t-1})$ **then** $\hat{y}_t = f(\hat{x}_t)$ for any $f \in \mathcal{F}_{t-1}$
    **else if** $\exists s_1, \ldots, s_\alpha < t$ *and* $\exists i_1, \ldots, i_\alpha \in [p]$ *such that* $\hat{x}_{s_j, i_j} \in [\hat{x}_{t,i_j}, a_{i_j}) \cup (b_{i_j}, \hat{x}_{t,i_j}]$ **then**
        $\hat{y}_t = 0$
    **else** $\hat{y}_t = \bot$
    Receiving label $y_t$
    Update $S_t \leftarrow S_{t-1} \cup \left\{(\hat{x}_t, y_t)\right\}$ and $\mathcal{F}_t \leftarrow \mathcal{F}_{t-1}^{\hat{x}_t \to y_t}$
    Update $a_1, \ldots, a_p$ and $b_1, \ldots, b_p$ such that $[a_1, b_1] \times \ldots [a_p, b_p]$ is the smallest rectangle
    containing the points labelled positive so far. i.e. $a_i = \min\{x_i : (x, 1) \in S_t\}$

---

If we have only seen 0 labels so far, the algorithm predicts 0. Otherwise, let $[a_1, b_1] \times [a_2, b_2] \times \ldots \times [a_p, b_p]$ be the minimal rectangle enclosing the positive examples so far (i.e., the Closure hypothesis). For the next point $\hat{x}_t$, if it isn't in the region of disagreement, we predict the agreed-upon label. Otherwise, we check whether there exist at least $\alpha$ examples $\hat{x}_s$, $s < t$, for each of which there exists

a coordinate $i$ with $\hat{x}_{si} \in [\hat{x}_{ti}, a_i) \cup (b_i, \hat{x}_{ti}]$ (at most one of these sides is non-empty – or sometimes both sides will be empty for some coordinates $i$). If so, we predict 0. Otherwise, we abstain. (The algorithm never predicts 1 in the region of disagreement, similar to the Closure algorithm).

*Proof of Theorem 5.4.* For any example with target label 1, if the algorithm predicts 0, there are at least $\alpha$ examples $\hat{x}_s$ which each have some coordinate $\hat{x}_{s,i}$ that was not in $[a_i, b_i]$ before the update, but which will have $\hat{x}_{s,i}$ in $[a_i, b_i]$ after the update. For each example $\hat{x}_s$, this can happen at most $p$ times (corresponding to each coordinate) before it will never again be included in a future set of $\alpha$ examples that convince us to predict 0. So we make at most $pT/\alpha$ misclassifications on adversarially injected examples. Since the algorithm never predicts 1 unless the true label is 1, we never misclassify a negative example. So it remains only to bound the number of abstentions on the i.i.d. examples.

For any $n$, suppose $\hat{x}_t$ is the $n$-th i.i.d. example, and let $\tilde{x}_1, ..., \tilde{x}_n$ be these $n$ i.i.d. examples (so $\tilde{x}_n = \hat{x}_t$). If $\hat{x}_t$ is "attackable" (same to Definition C.2 meaning that there is some set of examples the adversary could add, knowing $\hat{x}_t$, to make us abstain) then it must be that either $\hat{x}_t$ is a positive example in the region of disagreement of the version space induced by the other $n - 1$ points, or else $\hat{x}_t$ is a negative example such that there are $< \alpha$ points $\tilde{x}_s$, $s < n$, for which there exists $i$ with $\tilde{x}_{s,i} \in [\hat{x}_{t,i}, a_i) \cup (b_i, \hat{x}_{t,i}]$. In particular, in the latter case, it must be that each coordinate $i$ has $< \alpha$ examples $\tilde{x}_s$ with $\tilde{x}_{s,i} \in [\hat{x}_{t,i}, a_i^\star) \cup (b_i^\star, \hat{x}_{t,i}]$, where the target concept is $[a_1^\star, b_1^\star] \times \ldots \times [a_p^\star, b_p^\star]$. This is because the current estimated rectangle will be inside the true rectangle.

We will use exchangeability to bound the probability that $\tilde{x}_n$ is attackable by $\frac{1}{n}$ times the number of $\tilde{x}_s$, $s \leq n$, which would be attackable if they were swapped with $\tilde{x}_n$. Among $\tilde{x}_1, \ldots, \tilde{x}_n$ there are at most $2p$ positive examples in the region of disagreement of the version space induced by the others (namely, the minimum spanning set of the positive examples). For each coordinate $i$, there are at most $2\alpha$ examples $\tilde{x}_s$, $s \leq n$, with $< \alpha$ other examples $\tilde{x}_{s',i}$ in $[\tilde{x}_{s,i}, a_i^\star) \cup (b_i^\star, \tilde{x}_{s,i}]$ (namely, the $\leq \alpha$ examples with smallest $\tilde{x}_{s,i}$ such that $\tilde{x}_{s,i} > b_i^\star$, and the $\leq \alpha$ examples with largest $\tilde{x}_{s,i}$ such that $\tilde{x}_{s,i} < a_i^\star$). So there are at most $2\alpha p$ negative examples $\tilde{x}_s$ which would be attackable if they were swapped with $\tilde{x}_n$.

Altogether there are at most $2p(\alpha + 1)$ examples $\tilde{x}_s$ which would be attackable if they were swapped with $\tilde{x}_n$. Thus, the probability i.i.d. example $\hat{x}_t$ is attackable is at most $2p(\alpha + 1)/n$ where $n$ is the number of i.i.d. points seen so far including $\hat{x}_t$. Summing, the expected number of abstentions on i.i.d. examples is at most $2p(\alpha + 1) \log T$. Now setting $\alpha = \sqrt{T/\log T}$, gives us the desired result. $\qquad\square$

