# Adversarial Resilience in Sequential Prediction via Abstention

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

**Adversarial Examples.** Our clean-label attack is very closely related and motivated by the setting of adversarial examples [SZS+13, BCM+13, GSS14]. The goal in this setting is to learn a classifier that predicts correctly on all adversarial examples, which is a very strong requirement. Empirical work in this space has focused on designing methods to make training adversarially robust [MMS+17, WK18], and also on detecting adversarial examples [PDDZ18, AHFD22]. Detecting adversarial examples is a very challenging tasks and proposed solutions are often brittle [CW17]. In fact, our framework does not explicitly require detection as long as we can predict correctly on these.

## 2  Abstention Framework

In this section, we present the formal framework for sequential prediction with abstentions.

**Notation.** We will denote the domain with $\mathcal{X}$ and the distribution over $\mathcal{X}$ as $\mathfrak{D}$. We let $\Delta(\mathcal{X})$ denote the set of all distributions over $\mathcal{X}$. We will work in the realizable setting where our label will be according to some function in $\mathcal{F}$ with VC dimension $d$. Given a class $\mathcal{F}$ and a data set $S = \{(x_i, y_i)\}$, we will denote by $\mathcal{F}|_S$, the class $\mathcal{F}|_S = \{f \in \mathcal{F} : \forall i \quad f(x_i) = y_i\}$. When the data set contains a single point $S = \{(x, y)\}$, it will be convenient to denote $\mathcal{F}|_S$ as $\mathcal{F}^{x \to y}$.

**Protocol.** At the start, the adversary (or nature) picks a distribution $\mathfrak{D}$ over the domain $\mathcal{X}$ and the labelling function $f^\star \in \mathcal{F}$. We will be interested in both the setting where the learner knows the distribution $\mathfrak{D}$ and the setting where the learner does not know the distribution $\mathfrak{D}$. In the traditional sequential prediction framework, the learner sees input $x_t$ at time $t$ and makes a prediction $\hat{y}_t$ and observes the true label $y_t$. The main departure of our setting from this is that an adversary also

decides before any round whether to inject an arbitrary element of $\mathcal{X}$ (without seeing $x_t$). We denote by $\hat{x}_t$ the instance after the adversarial injection ($\hat{x}_t = x_t$ or $\hat{x}_t \neq x_t$). The learner then observes $\hat{x}_t$ and makes a prediction $\hat{y}_t$ and observes the true label $y_t$, as in the traditional sequential prediction framework. We present this formally as a protocol in 2.

---

**Protocol 1** Sequential Prediction with Adversarial Injections and Abstentions

---

Adversary (or nature) initially selects distribution $\mathfrak{D} \in \Delta(\mathcal{X})$ and $f^\star \in \mathcal{F}$. The learner does not have access to $f^\star$. The learner may or may not have access to $\mathfrak{D}$.

**for** $t = 1, \ldots, T$ **do**

    Adversary decides whether to inject an adversarial input in this the round ($c_t = 1$) or not ($c_t = 0$).

    **if** $c_t = 1$ **then** Adversary selects any $\hat{x}_t \in \mathcal{X}$

    **else** Nature selects $\hat{x}_t \sim \mathfrak{D}$

    Learner receives $x_t$ and outputs $\hat{y}_t \in \{0, 1, \perp\}$ where $\perp$ implies that the learner abstains.

    Learner receives clean label $y_t = f^\star(\hat{x}_t)$.

---

It is important to note we are in the realizable setting, even after the adversarial injections since the labels are always consistent with a hypothesis $f^* \in \mathcal{F}$. This model can naturally be extended to the agnostic setting with adversarial labels.

**Objective.** The goal of the learner is to have low error rate on the rounds they decides to predict (that is when $\hat{y}_t \in \{0, 1\}$) while also ensuring that they do not abstain ($\hat{y}_t = \perp$) on too many non-adversarial rounds ($c_t = 0$). More formally, the learner's objective is to minimize the following error (or regret),

$$\text{Error} := \underbrace{\sum_{t=1}^{T} \mathbb{1}[\hat{y}_t = 1 - f^\star(x_t)]}_{\text{MisclassificationError}} + \underbrace{\sum_{t=1}^{T} \mathbb{1}[c_t = 0 \wedge \hat{y}_t = \perp]}_{\text{AbstentionError}}.$$

We could formulate a relaxed cost-based version of this objective that allows us to trade-off these errors, however, we will focus on the strong notion of error for this paper.

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

- In the unknown distribution case, extending our result for VC dimension $d > 1$ classes is wide open. Though, we have algorithms for certain classes of higher VC dimension, such as intervals and axis-aligned rectangles, they seem to heavily exploit the structure of the particular class. Thus, showing either an upper or lower bounds of the error dependence on both the class and the time horizon would be interesting.
- The unknown distribution case can be seen as form of distribution-free uncertainty quantification. It would be interesting to understand connections to other forms such as conformal prediction. On a technical level, our work exploits exchangeability of the i.i.d. sequence which is the foundation of conformal prediction, though the main challenge in our setting is the presence of adversarial inputs. It would be interesting to build on this connections and understand whether techniques can be ported over in either direction.
- Our focus for this paper has been entirely on the statistical benefits of abstention. Understanding the computational complexity in this setting is an exciting avenue for research. Concretely, for halfspaces in $d$ dimensions, is there a polynomial time algorithm for learning with abstentions, even for well-behaved distributions such as Gaussians. On a related note, showing computational-statistical gaps in this specialized setting would be interesting, albeit disappointing.

**Broader Impact.** The aim of our work is design a theoretical framework for handling adversarial examples and guaranteeing certainty in sequential prediction. Our model assumes no cost for abstaining on adversarial inputs. However, in real-world scenarios, these costs can be high, consider the human cost of evaluating each of the abstained examples. Furthermore, as with any framework, application of our metric without taking into account the nuances of the real-world situation could lead to unintended consequences, for example, a high abstention rate on marginalized sub-populations.

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

## C   Discussion

In this section, we further discuss the proposed algorithms and propose potential ways to extend these beyond the current settings.

### C.1   Generalizing Higher-order Disagreement to Unknown Distribution

In Section 4, we saw an algorithm that for any class $\mathcal{F}$ with VC dimension $d$, achieves abstention error and misclassification error bounded only as a function of $d$. As mentioned earlier, an interesting open question is to extend this to the unknown distribution case. In this section, we will briefly discuss a natural algorithm extending the algorithm from the known distribution case. Recall that Algorithm 1 computed the probabilities of shattering $k$ points using the knowledge of the distribution and made a prediction $\hat{y}_t$ depending on the relative magnitudes of the probabilities corresponding to the two restricted classes. The main challenge in the unknown distribution case is that it is not immediately obvious how to compute these quantities.

One natural approach is to use the historical data as a proxy for the distribution. That is, given the data set $S_t$ of size $n$, compute the leave-$k$-out estimator for the probability as follows

$$\tilde{\rho}_k(S, \mathcal{F}) = \frac{1}{\binom{n}{k}} \sum_{T \subset S; |T|=k} \mathbb{1}\left[T \text{ is shattered by } \mathcal{F}|_{S \setminus T}\right].$$

There are a few things to observe about this estimator. First, in the case when the data is generated i.i.d., this estimator is unbiased. Further, though each of the summands is not independent, one can show concentration for estimators of this form using the theory of U-statistics. Additionally, recall that in Algorithm 1 required the thresholds $\alpha_k$ to be set to $T^{-O(k)}$ (we use $T^{-k}$ but it is straightforward to extend this to $T^{-ck}$ for $c < 1$). This appears to be high precision but note that the "number of samples" one has for a data set of size $n$ is $n^{O(k)}$. Thus, it is conceivable that such an estimator can give the necessary bounds. Unfortunately, the challenge with analyzing this in our setting is that we do no know which of the examples are corrupted. Thus, the adversary could inject examples to make our estimates arbitrarily inaccurate. Thus, as in the case of the VC dimension 1 classes we saw in Section 5 and the case of the rectangles we will see subsequently, our analysis would need to not rely on the accuracy of the estimates but rather use these estimates to maintain progress or construct other versions of estimators that are unaffected by the adversarial injections.

### C.2   Generalizing Structure-based Algorithm beyond VC Dimension 1 Classes

In Section 5 we saw an algorithm that for any class $\mathcal{F}$ with VC dimension 1, achieves abstention and misclassification error bounded by $O(\sqrt{T})$ without access to the underlying distribution. An interesting open question is to extend structure-based algorithms beyond the VC 1 classes. Here we will show that for the class of axis-aligned rectangles in dimension $p$ (VC dimesnion is $2p$), we can indeed design an algorithm that achieves abstention and misclassification error bounded by $O(p\sqrt{T \log T})$. This exhibits a class of VC dimension $> 1$ for which we can attain the desired guarantees without access to the distribution.

Both our algorithms heavily utilize the structure of the underlying function class and analyze based on the notion of attackability. It would be interesting to characterize other structural hypothesis classes that enjoy similar guarantees. A natural extention to the axis-aligned rectangles would be any intersection closed hypothesis class.