# OpenReview forum: "Adversarial Resilience in Sequential Prediction via Abstention"
_NeurIPS.cc/2023/Conference — NeurIPS 2023 poster_

### Official Review · Reviewer_mVCw · 2023-06-20

**Soundness:** 3 good
**Presentation:** 3 good
**Contribution:** 3 good
**Rating:** 6
**Confidence:** 3

**Summary:**

The authors study online learning under clean-label attacks. Since it is online learning, such attacks can be seen both as poisoning as well as evasion (adversarial examples). In this direction the authors propose the use of abstention when the classifier is not confident for a prediction and along these lines the main contribution is the introduction of a new loss function that penalizes abstention on non-adversarial examples but not on adversarial examples. The authors give an error (regret) bound of $O(d^2\log T)$ ) for classes with VC dimension d for time horizon T, assuming access to the marginal distribution over the iid examples. This bound is off by a factor of $d$ from the situation where it is known that data are coming iid from some fixed distribution but at the same time the authors allow clean-label attacks which essentially alter the distribution. Finally, the authors also give a $O(\sqrt{T})$ bound for concept classes of VC dimension 1 but without any access to the marginal distribution over the iid examples.

After the rebuttal:

We had an interesting discussion during the rebuttal period and I am happy to see the authors will be integrating the feedback from the reviewers and improve the final version of the paper. Therefore I am increasing the score on soundness and presentation from 2 to 3 and the overall rating of the paper from 3 (reject) to 6 (weak accept). Thank you for a very interesting paper!

**Strengths:**

The paper is studying sequential prediction while the training examples can be poisoned in the sense that they are not drawn iid from some underlying distribution but otherwise respect the ground truth. This model of learning is somewhere between traditional machine learning where the learner has access to iid examples and online learning where the learners are studied under worst-case sequences of examples that are presented to the learners. The authors study the situation of sequential prediction, which is a term hardly found anywhere online and they never define in the text, but apparently has to do with the fact of studying machine learning algorithms that make predictions as they still learn from data (e.g., perceptron); the catch is to what extent the sequence of data can be adversarial and in this sense the authors allow perhaps the most adversarial of sequences. I believe this is an interesting problem though it is unclear how important this problem is, if it arises in the actual world, and where the motivation is coming from. The paper is sometimes well-written and sometimes not. Hence, there are issues both with the significance of the results and with the clarity.

I like and appreciate the concluding section with ideas for future work and a clearly-separated paragraph on broader impact. However, the authors do not have a similarly marked paragraph listing the limitations of their work.

**Weaknesses:**

I think the paper has good potential, but it is not quite polished yet.

Several definitions are missing or are confusing in the paper. Examples follow:

- Sequential prediction is never defined as a concept.

- Learning in the realizable case vs proper learning: in line 28 you define the hypothesis class $F$ and in lines 133-134 you indicate that labeling occurs according to some function in $F$. This is actually proper learning, which is a special case of learning in the realizable case.

- In line 135 you forget to mention that you define what a version space is (and cite Mitchell) but you go ahead and use this notion in lines 255 and 269 (and perhaps elsewhere).

- Littlestone dimension is mentioned but is never defined (not even in the appendix) and no example is actually given.

- In lines 186-190: What is the relationship between $n$ and $T$? Does $n$ refer to unique examples, or does it allow potential repetitions and is thus the full number of draws from the (potentially adversarial) distribution?

- One of the main contributions of the paper is the new loss function that penalizes the learning algorithm when when it abstains on non-adversarial data but does not penalize the learner on adversarial data. This is captured with the equation on display in line 152 where we see that the total error that the learner suffers over the sequence of the first T examples is decomposed to misclassification error and abstention error. Since the characteristic function (I assume that is what 1{$\ldots$} is) is never defined, I understand from the misclassification error that the learning algorithm should receive a penalty of 1 every time it abstains since $\hat{y}_t \neq 1 - f^*(x_t)$. However, in Proposition 3.1 the learning algorithm abstains to predict in the disagreement region and the claim is that it has misclassification error equal to 0. However, I have no idea how this follows from the equation that is put on display in line 152.

- In Protocol 1, two lines before the end you claim that the learner receives $x_t$ but earlier, in lines 143-144 it is clear that the learner receives $\hat{x}_t$.

Near the end the paper becomes more dense and while I appreciate that the authors are attempting to enhance the presentation by leaving the proofs in the appendix (thank you) there are still situations where there is ambiguity (e.g., in Theorem 5.1, is the "corruption model with abstentions" something new?)

In line 248 "As we saw earlier..." -> Perhaps explicitly number the particular example and refer to that with a number?

As a last remark, I am not sure if NeurIPS is the appropriate venue for this line of work. I do not believe it is.

**Questions:**

Q1. What is the relevance of adversarial examples and poisoning attacks to medical applications? How is this line of work motivated for both kinds of attacks?

Q2. Going one step further, what is the motivation for studying such methods as the ones you analyze in the paper?

Q3. Can the functions in $F$ be randomized?

Q4. What is the difference between the functions 1{$\ldots$} (line 152) and I{$\ldots$} in lines 295 and 301?

Q5. Is there an updated version of the loss function that you propose?

Q6. Line 317: "... the class is restricted ...". What class?

Q7. How tight are the bounds with respect to some actual datasets that motivate this line of work (e.g., from the medical domain that is mentioned)?

**Limitations:**

I would prefer to see a clearly separated paragraph somewhere near the end of the paper indicating clearly what the limitations of the proposed approach are as well as to see some experiments with real data that show how close the bounds are to reality in situations that have motivated this line of research.

---

> ### Author Rebuttal · Authors · 2023-08-09
>
> We thank the reviewer for the review. We will promptly fix typographical errors in a revision. Here we address the major concerns/questions from the reviewer.
>
> **Proper versus realizable**: Realizability and properness are somewhat orthogonal desiderata in learning theory. Realizability is a requirement on _learning instances_ and implies that there is a function in the class that exactly fits the labels from the training set (potentially presented in a sequential manner). Properness is required from the _learning algorithm_ and implies that the algorithm uses a function from the class to make its predictions. These two requirements do not enforce each other in any way. For example, optimal learning algorithms even in the realizable setting may need to be improper and there are settings, such as learning in games, where the instance is not realizable yet a proper algorithm is desired.
>
> **Definitions and notation**: We apologize for this omission of definition of some terms such as sequential prediction and version space that we believe are standard nomenclature in learning theory. We will include formal definitions to improve readability.
>
> **Relationship between $n$ and $T$**: In that paragraph, $n$ is used to refer to the number of samples that are not corrupted while $T$ is used to refer to the total number of samples. In general, $n \leq T$ (could take any value in $\lbrace 0, 1, \ldots, T\rbrace$) but the algorithm does not have access to $n$ and we can only use it in the analysis. We also comment that the i.i.d. samples could in fact have repetitions.
>
> **Notion of error**: We recall that we use the characteristic function in the standard way i.e. to be the function that outputs one when the condition in the input is satisfied and zero when the condition is not satisfied. Note that the notion of misclassification error is $\sum_i 1( \hat{y_t} = 1 - f(x_t) )$. This means that the algorithm suffers an error of one exactly when it does **not** abstain and predict the opposite label. For example, when $\hat{y_t} = \bot$ the algorithm does not suffer a misclassification error as $\hat{y_t} \neq 1 - f(x_t)$. In particular, when the algorithm abstains in the disagreement region, it does not make a prediction and does not suffer a misclassification error.
>
> **Limitations**: Thank you for your suggestion, we will add a clearly marked paragraph highlighting the limitations of our current model and approaches.
>
> **Appropriateness for Neurips**: We strongly disagree with this. Just as evidence for this, related paper working on similar setups (generally on beyond worst case analysis of sequential prediction) have appears in previous editions of NeurIPS:
> - Beyond Perturbations: Learning Guarantees with Arbitrary Adversarial Test Examples NeurIPS 2020 (oral)
> - On optimal learning under targeted data poisoning NeurIPS 2022
> - Oracle-eﬃcient online learning for beyond worst-case adversaries NeurIPS 2022 (oral)
> - Towards optimally abstaining from prediction with OOD test examples NeurIPS 2020
>
> **Responses to Questions**:
>
> Q1&2. As we mentioned in the introduction, our model captures both data poisoning and OOD example. The key benefit of our model and algorithms is the fact that they produce 'certain' predictions. In particular, the model is mostly correct whenever it predicts despite out-of-distribution or adversarial data. Creating predictors equipped with this guarantee is very desirable in high-stakes applications where incorrect predictions are costly, and we need high-levels of reliability from our models. As for the medical scenario, the OOD setting is more natural. For example, if our model has seen data mostly from a particular population (say the majority population), then our model is likely to make incorrect predictions on patients not from this population. In such scenarios, it is better for the predictor to abstain than make an incorrect prediction on the minority population. Similar OOD interactions can occur in other settings such as self-driving cars, chatbots, etc.
>
> Q3. We assume that the functions in $F$ are deterministic functions. But since the learning occurs in a sequential manner, the adversary can pick functions (in potentially a randomized manner) as long as they are consistent with the data (that is, the function matches the labels seen so far). One can also incorporate functions such that the label is random given the input but this is usually studied in the agnostic setting (not realizable). Most results in learning theory are first proven for the realizable case and the general agnostic case is solved via reduction to the realizable one. Our paper can be seen as putting forth the first step.
>
> Q4. We use both $1$ and $I$ to both mean the indicator function. We will fix this in the revision.
>
> Q5. We propose a loss corresponding to our setting given by the equation in line 152. But, we believe that thinking of our setting directly in terms of loss minimization is not fruitful. Since we do not have access to whether the examples the learner has seen are adversarially injected or i.i.d. ($c_t$ is unknown), the algorithm cannot compute its own loss (in particular, the learner cannot compute the abstention error). This highlights another major challenge of our setting.
>
> Q6. Here by the class, we mean the class with respect which we are learning, referred to as $\mathcal{F}$ in the paper
>
> Q7. The focus of our work is mainly theoretical. Our goal in this paper was to propose a model under which sets a realistic goal on handling adversarial/OOD data and showing the existence of efficient algorithms in this setting. We hope that our work will inspire practitioners to develop practical algorithms using the algorithmic ideas proposed in this work. For example, our uncertainty estimation measures based on shattered sets can be used to create heuristics that work for classes such as neural networks.

---

> > ### Comment · Reviewer_mVCw · 2023-08-15
> > **Thank you for the response**
> >
> > Thank you for the response. I understand that the work has merit, but even after the response that the authors gave to the issues raised by the various reviewers, I think that the paper is not ready for acceptance yet. I have some minor follow-up comments to your comments.
> >
> > **Proper versus realizable.** Proper learning and realizability are not orthogonal issues. Whenever one uses a proper learning algorithm, indeed they learn in the realizable setting.  However, it is not necessarily true that when one has a learning algorithm that satisfies the realizability assumption, that we do proper learning. And indeed, representation-independent learning that satisfies the realizability assumption may provide more power to the learner; there are classic results along these lines as well as recent ones.
> >
> > One formally needs to distinguish between the hypothesis class and the concept class, which you do not really do in your paper. Having said that you do say that labeling happens according to some function in $\mathcal{F}$ (lines 133-134) and moreover the model that you learn will be one of the functions in $\mathcal{F}$ -- that is your *hypothesis class* anyway. This *is* the definition of *proper learning* because what you are saying is that the hypothesis class coincides with the concept class, since the labeling of the ground truth function happens according to some function in the *hypothesis class*.
> >
> > **Appropriateness to NeurIPS.** Among the papers that you mention, only one of them is really about online learning.  Having said that, even if you can bring one (or few) papers out of thousands that get accepted every year does not mean you have a strong point. Furthermore, continual learning for neural networks leads to catastrophic forgetting.
> >
> > **Answers to my questions.** Thank you for the answers to my questions. Since there is motivation from the medical domain (or even other domains) and you think that the paper is relevant to NeurIPS, I would like to see some experimental results from at least one of the domains that motivate this line of work, so that we can understand better the relationship between the theoretical results and the practical aspects of the proposed methods.

---

> > > ### Author Response · Authors · 2023-08-16
> > >
> > > **Proper versus realizable**: Proper and realizable are indeed orthogonal concepts. The statement “_Whenever one uses a proper learning algorithm, indeed they learn in the realizable setting_” is not necessarily true. To be in the realizable setting, the **label** in the data distribution needs to be exactly according to a function $f^* \in \mathcal{F}$ which implies there is no noise in the data (with respect to this function class). While proper learning only asks for the function used to make predictions by the learner to be in $\mathcal{F}$. Your statement implies that whenever we desire our learner to be proper, the data is not allowed any noise, which is not necessary. There are many non-realizable settings (such as learning in games) where one might want to be proper.
> > >
> > > As you point out, we do state that we are in the realizable model, that is, our labels are always consistent with some $f^* \in \mathcal{F}$ but we do not restrict our algorithm to predict according to a function in the class.
> > > In particular note that our algorithm decides the labels *after* looking at the point $x_t$ and no constraints are imposed on the choice of $\hat{y_t}$. So the final model we learn will not necessarily be in the class. In fact, since we are in the online setting (additionally with abstentions) and learner can decide to change the prediction function in a time dependent manner, the term *final model* needs to be interpreted carefully. Our algorithm or its analysis does not imply that we recover the original function. In fact our model is designed to smoothly interpolate between settings where learning is *impossible* and settings where learning is possible.
> > >
> > > We are happy to add a clarification about our algorithms being improper in the paper.
> > >
> > >
> > > **Appropriateness to NeurIPS**: We only gave a few examples of papers that are most closely related to our work that appeared in the recent NeurIPS, two of which appeared as orals. We could produce a much longer list if needed of theoretical work on online learning, since online learning and learning theory are fairly well represented at NeurIPS. Perhaps the strongest point we can make is that the ‘Call for papers’ for NeurIPS 2023 clearly states ‘online learning’ as a topic of interest, see bullet “General machine learning (supervised, unsupervised, **online**, active, etc.)” along with "Theory (e.g., control theory, **learning theory**, algorithmic game theory)".
> > >
> > > **Catastrophic forgetting**: Note that our setting assumes that the true labeling function remains the same throughout time, and our guarantees ensure that we will not make too many errors on data drawn from the fixed distribution $\mathcal{D}$ throughout the sequence, therefore we do not need to worry about catastrophic forgetting. The goal of the learner is to make certain predictions on the i.i.d. data while handling adversarial/OOD data. Note that our total error is bounded by $o(T)$, hence the fraction of errors over time goes to 0 as $T \rightarrow \infty$, as is standard in online learning.
> > >
> > >
> > > **Experimental evaluation**: We strongly disagree with the reviewer on their claim that “_Since there is motivation from the medical domain (or even other domains) and you think that the paper is relevant to NeurIPS, I would like to see some experimental results from at least one of the domains that motivate this line of work_”. Note that most theoretical work in machine learning is motivated by a real world problem, therefore it seems like the reviewer is claiming that for a paper to be appropriate for NeurIPS, experiments are mandatory. Our work poses a new paradigm for beyond worst-case learning in the online learning framework which is a very important problem in the field of online learning (see our related work). We subsequently give new algorithmic results showing that adversarial data can be handled without paying much overhead compared to the completely non-adversarial setting. We believe our work will inspire a lot of subsequent work in the ML community (as evident from the numerous extensions possible of our framework and connections to several different problems actively studied by the community including uncertainty estimation, conformal prediction, testable learning, and adversarial examples). We value the importance of empirical evaluations in general, however we do not believe experiments are essential to our paper. Our paper's main focus is purely theoretical and that is why we chose the category of 'learning theory', which our paper fits well in.
> > >
> > > The reviewer says that ‘_I think that the paper is not ready for acceptance yet_’. **Could the reviewer please give us concrete reasons for this concern?**

---

> > > > ### Comment · Reviewer_mVCw · 2023-08-16
> > > > **Brief comment**
> > > >
> > > > For the proper learning and the realizable case, I agree with you that you have not done a good job in explaining these things well in your paper and this needs clarification from your end in the paper.  That's one thing.
> > > >
> > > > Another thing is that you are using a straw man argument to avoid the discussion of the actual point.  Let me first answer the straw man argument that you have: In the absence of additional information, comparing proper learning and learning in the realizable case, one in fact discusses what I described above; namely when one does proper learning, then they also work in the realizable case. That is one thing.  The other thing is, that indeed, distinguishing between the concept class and the hypothesis class would help a lot to clarify your findings in the paper (after all, you do know that there is such a thing as a *concept class*, since in line 115 you refer to the target *concept*).
> > > >
> > > > However, the above discussion is unrelated to the actual point of my initial comment.  Namely:
> > > >
> > > > - in lines 21 and 28 (and perhaps elsewhere too) of your paper you clarify to the reader that the set of functions $\mathcal{F}$ is your *hypothesis class*, and
> > > >
> > > > - at the same time you also explain in line 138 that one of the functions in $\mathcal{F}$ does the labeling; i.e., the target concept is one of the functions in $\mathcal{F}$.
> > > >
> > > > And I, as a reader, am using the combined knowledge of these two statements to note that these two pieces of information in fact imply *proper learning*. In your last response, you accept that the above two statements are indeed referring to proper learning, but somehow this time you want to argue that $\mathcal{F}$ is not actually your hypothesis class after all, contradicting your claims in the paper.
> > > >
> > > > Of course, you have every right to change your opinion on what the hypothesis class is.  However, this thing alone makes it sort-of self-evident that _even though you have a very nice story to tell_, it is still to some extent convoluted so that the readers can understand and appreciate your results and put them in the right perspective. Please make the setting and the learning mechanism clearer and I am positive that this paper will have better chances of being accepted even without experiments from which you want to steer clear for some reason.
> > > >
> > > > Also, regarding the experiments, theoretical work in NeurIPS, is typically backed up by experiments. It appears that you want to argue only about theory and you definitely want to steer clear from experiments. If you are so dismissive of doing experiments with data from *any* field among the several that you use as argument for motivating your work, then one can only wonder how good these results actually are and this is on top of an unclear presentation of some topics in your paper. There are also purely theoretical venues such as COLT and ALT where this paper could fit better?
> > > >
> > > > As a concluding summary: I believe that you have some interesting results and story to tell in a revised version of this paper, but this paper cannot receive my vote of confidence for acceptance in its current form.

---

> > > > > ### Comment · Reviewer_mVCw · 2023-08-17
> > > > >
> > > > > And to further clarify misuse of terms, in your last comment you state:
> > > > >
> > > > > _we do state that we are in the realizable model, that is, our labels are always consistent with some $f^*\in\mathcal{F}$ but we do not restrict our algorithm to predict according to a function in the class. In particular note that our algorithm decides the labels after looking at the point $x_t$ and no constraints are imposed on the choice of $\hat{y_t}$. So the final model we learn will not necessarily be in the class._
> > > > >
> > > > > Let's look at what really a _hypothesis class_ is:
> > > > >
> > > > > - Understanding Machine Learning: From Theory to Algorithms (https://www.cs.huji.ac.il/~shais/UnderstandingMachineLearning/understanding-machine-learning-theory-algorithms.pdf),
> > > > > 4 lines before the end of page 36 the definition is given for the _hypothesis class_:
> > > > >
> > > > >   _Formally, the learner should choose in advance (before seeing the data) a set of predictors. This set is called a _hypothesis class_ and is denoted by $\mathcal{H}$. Each $h\in\mathcal{H}$ is a function mapping from $\mathcal{X}$ to $\mathcal{Y}$._
> > > > >
> > > > >   .
> > > > >
> > > > >   That is, the fact that you come up with a hypothesis outside of the hypothesis class, defies the expected use of the term _hypothesis class_.  And it is in this context that you further say that the ground truth function is selected from this set of functions and moreover that the realizability assumption holds. And in my comment that this is in fact _proper learning_ you come up with discussion about noise that is irrelevant to our discussion, pretending that you win some argument.
> > > > >
> > > > > This definition is not just from the above book.  Let's look at another very recent and equally good book:
> > > > >
> > > > > - Foundations of Machine Learning, second edition (https://www.dropbox.com/s/38p0j6ds5q9c8oe/10290.pdf?dl=1), Section 1.4, near the very top of page 5:
> > > > >
> > > > >   _Hypothesis set: A set of functions mapping features (feature vectors) to the set of labels $\mathcal{Y}$._
> > > > >
> > > > >   Near the end of that page it is also stated for clarification to the readers:
> > > > >
> > > > >   _the algorithm selects a different hypothesis out of the hypothesis set._
> > > > >
> > > > >   .
> > > > >
> > > > >   The authors here use the term of _hypothesis set_ for what was earlier called the _hypothesis class_ (as in your paper). Again, it is crystal clear that this set of functions is the set of functions where the learner is allowed to select one of them as the final _hypothesis_ (that is: a _guess_ on how the ground truth looks like; words have not been chosen by chance)
> > > > >
> > > > >
> > > > > I can of course give you more examples of the same definition from other books that agree with my point of view and not yours.
> > > > >
> > > > > As a summary, people can use the terms _hypothesis class_, _hypothesis set_, and _hypothesis space_, for referring to the set of functions from where the learner chooses one function and uses it for prediction during the test time. If it is further true that the ground truth function can be _any one_ of the functions in this set, then it is the case that the hypothesis class _coincides_ with the concept class and we are talking about _proper learning_.

---

> > > > > > ### Author Response · Authors · 2023-08-19
> > > > > >
> > > > > > Thanks for taking the time to engage with our paper. We acknowledge the confusion caused by our use of the terms concept class and hypothesis class and we apologize for this. $\mathcal{F}$ is intended to be the concept class. We will correct this in a revision. That said, we still believe that this does not affect any of the technical claims and contributions of the paper.
> > > > > >
> > > > > > Furthermore, we don't not believe that experiments do not add to even otherwise theoretical papers. It is just that the main point that we wanted to convey was that, like most work in learning theory, while our work is inspired by real world considerations, direct experiments do not capture the conceptual contribution made by the paper. The aim of our paper is to present a new theoretical framework and general algorithms that highlight that it is possible to achieve stochastic like guarantees even in the presence of adversarial data by using the option of abstention wisely. We hope that more empirical papers can build upon our framework and algorithms to make them more efficient using domain specific information.

---

> > > > > > > ### Comment · Reviewer_mVCw · 2023-08-19
> > > > > > >
> > > > > > > Assuming that today my comments can indeed go through, let me come to another issue.
> > > > > > >
> > > > > > > Since we agree now that $\mathcal{F}$ is your concept class, then the realizability assumption as stated also has issues.  The reason is that you claim in the text (lines 133-134), as well as in a comment above, that because the labels respect the ground truth $f^*$, then the realizability assumption holds. By itself, this statement is completely wrong, especially in the context of the arguments that you wanted to have earlier when you were comparing learning in the realizable case versus proper learning.
> > > > > > >
> > > > > > > More information about realizability:
> > > > > > >
> > > > > > > - [B1, Definition 2.1, page 38]
> > > > > > >
> > > > > > > - [B2, bottom of page 29]
> > > > > > >
> > > > > > > [B1] Understanding Machine Learning: From Theory to Algorithms
> > > > > > > https://www.cs.huji.ac.il/~shais/UnderstandingMachineLearning/understanding-machine-learning-theory-algorithms.pdf
> > > > > > >
> > > > > > > [B2] Foundations of Machine Learning, second edition
> > > > > > > https://www.dropbox.com/s/38p0j6ds5q9c8oe/10290.pdf?dl=1
> > > > > > >
> > > > > > > Some comment?

---

> > > > > > > > ### Author Response · Authors · 2023-08-19
> > > > > > > >
> > > > > > > > I am not sure I understand. But for the sake of clarity (in this conversation), let us put aside the issues of terminology for a second. In our setting there is class of function $\mathcal{F}$ (again let us not focus on what this class is called). The learner knows this class $\mathcal{F}$. Further, there is a $f^*\in \mathcal{F}$ that is unknown to the learner. The data is generated as $\tilde{x_t}$ (through the process described in the paper). Then a label is generated as $f^*(\tilde{x_t})$. This is exactly all we require in our analysis. Is there something wrong with this assumption?
> > > > > > > > In particular, this assumption implies both (a) there exists a hypothesis that has zero misclassification error on the historical data (b) there is a labelling policy (unknown to the learner) that suffers zero misclassification and abstention error on future data.
> > > > > > > > This is the sense in which we use the term realizability.

---

> > > > > > > > > ### Comment · Reviewer_mVCw · 2023-08-19
> > > > > > > > >
> > > > > > > > > Dear Author,
> > > > > > > > >
> > > > > > > > > When one generates the data respecting the output of the ground truth $f^*$, only says that the data is not affected by noise (for a deterministic $f^*$).  However, this is not enough so that you can claim realizability. In fact, none of the papers that you mentioned above, nor classic results from learning theory that rely on realizability (e.g., simple results such as Occam's razor from the '80s), see realizability in this way. In fact, if I understand correctly, not even this paper that we are discussing is using realizability with this definition.
> > > > > > > > >
> > > > > > > > > Realizability gives sufficient power to the learner to guarantee the existence of at least one function (called *hypothesis*) that classifies the training examples with no mistakes.
> > > > > > > > >
> > > > > > > > > To see the issue of your current definition, consider the hypothesis class that has only two constant functions: $\mathcal{H} = ${$h_1 = true, h_2 = false$}. Now, if you pick some function $f^*$ (other than what we find in $\mathcal{H}$) from some learnable concept class $\mathcal{C}$ and generate examples such that the labels are indeed the legitimate labels of $f^*$, then we can clearly create sequences of training examples where all the labels are indeed the output of $f^*$ but no $h\in\mathcal{H}$ can make zero mistakes in the entire sequence (we only need 1-2 examples and we are done). That is what realizability is about.
> > > > > > > > >
> > > > > > > > > In other words, it is not the case that when we assume realizability, it is the same thing as saying that the labels in our training examples are consistent with the output of some ground truth function $f^*$ (the claim of the paper). Realizability gives power to the learner. If we wanted to put a set inclusion between the hypothesis class $\mathcal{H}$ and the concept class $\mathcal{C}$, we would write down $\mathcal{H} \supseteq \mathcal{C}$. (And this set inclusion is why, ultimately, such a discussion is not orthogonal to the discussion about proper learning where we have $\mathcal{H} = \mathcal{C}$.)
> > > > > > > > >
> > > > > > > > > Is it clearer now? Do you disagree with anything from the above?

---

> > > > > > > > > > ### Author Response · Authors · 2023-08-20
> > > > > > > > > >
> > > > > > > > > > We think the confusion of the reviewer is arising from the misunderstanding of the online learning framework (in contrast to the offline/batch learning framework). We refer the reader to Chapter 21 of [B1] (from the reviewer's references). In online learning/sequential prediction, we do not restrict the prediction to be according to a "hypothesis class" but allow $y_t\in \{0,1\}$ (without any restrictions) at any time $t$. The realizability (see Section 21.1 in [B1]) is defined with respect to a fixed function class (you may call it concept class, the book refers to this as the hypothesis class as in our paper) which the learner has access to. Since predictions are allowed to be arbitrary, your example does not hold here, the learner can predict with respect to any boolean function at each time $t$ including $f^*$.
> > > > > > > > > >
> > > > > > > > > > If the reviewer examines the precise formalism of our model (Protocol 1) presented in the paper, they will find that it clearly states the conditions on the label and the predictions of the learner. Furthermore, our technical results are completely compatible with the definitions of our model. Therefore, we would like to shift the focus of the conversation away from a technical detail that has minimal relevance to our paper, and towards the reviewer's evaluation of the technical merit of our paper. We have resolved a majority of the reviewer's concerns and kindly urge the reviewer to reflect this in their score.

---

> > > > > > > > > > > ### Comment · Reviewer_mVCw · 2023-08-20
> > > > > > > > > > >
> > > > > > > > > > > The book calls the class hypothesis class as it should be called and this also what I am saying. If you actually read that portion, the learnt model is a function from that set of functions, which I am asking you to call hypothesis class. You don't.  In fact, above you explicitly explained that your final hypothesis can be outside of the hypothesis class, which is absurd. This never happens in the book.
> > > > > > > > > > >
> > > > > > > > > > > Notice also that the way the book defines realizability is by indeed having a *hypothesis* that can predict the labels exactly. Your paper has the claim that you work under the realizability assumption because the labels obey the output of the ground truth. Do you understand the difference?  It is one thing for the labels to be the output of the ground truth (i.e., no noise) and another thing for the learner to have the ability create a hypothesis that makes no mistakes in the sequence of examples provided (no matter what that sequence is). That is what realizability is about and this is exactly what once again this book stresses in Chapter 21. The book also focuses only on $\mathcal{H}$ because all computational results are with respect to the hypothesis class; not the concept class (be it online learning or batch learning). The learner needs to search among functions in $\mathcal{H}$ and that is where the computational effort is being put so that one can identify a good hypothesis.
> > > > > > > > > > >
> > > > > > > > > > > Do you think you can add a few lines and explain your framework and the relationship of your final hypotheses with the ground truth  a little bit better in the paper in the beginning of the paper, so that the reader is better prepared to understand what you will be doing and in fact see that realizability is used once again in the standard way (as you actually do)?  I am not shifting the focus away from anything. The difference is very fundamental and it makes it hard for anyone to understand what your model is and what you are actually doing.
> > > > > > > > > > >
> > > > > > > > > > > Finally, in case it is helpful, I believe that in your case, you have the relationship $\mathcal{H} \supseteq \mathcal{F}$, where $\mathcal{F}$ is the concept class and $\mathcal{H}$ is the hypothesis class (so that your final hypothesis belongs to what is traditionally called hypothesis class), because if I understand correctly, you allow all the functions in $\mathcal{F}$ (from where the labelling function $f^*$ is coming from and has no abstention; that's the concept class), as well as different versions of such functions where you allow abstentions for subsets of the domain and this combined set of functions is your $\mathcal{H}$. This is clearly a case that is realizable, but you should probably know better than me how to describe the relationship between these two sets and if this is everything.

---

> > > > > > > > > > > > ### Comment · Reviewer_mVCw · 2023-08-20
> > > > > > > > > > > >
> > > > > > > > > > > > Here is an alternate idea.
> > > > > > > > > > > >
> > > > > > > > > > > > It is probably easiest to indeed leave $\mathcal{F}$ as your hypothesis class as you have it in the paper.  This way you will not have to change statements in other parts of the paper, but then you need to clarify that $f^*$ is a legitimate hypothesis that has no abstentions anywhere in the domain and $\mathcal{F}$ contains such functions as well as more that are duplicate of these but allow abstentions in subsets of the domain because ultimately that is your goal?  If this is correct interpretation of $\mathcal{F}$, then this is indeed learning in the realizable case, not proper learning, and most likely leaves the entire paper as is.
> > > > > > > > > > > >
> > > > > > > > > > > > However, I think that some clarification is needed along these lines in the Protocol section where you explain that $\mathcal{F}$ is the set of functions (potentially partial) that the learner is allowed to output and a comparison of these functions with their counterparts where no abstentions exist. Such a comment would clarify the use of the term hypothesis class for $\mathcal{F}$ and would also explain that $f^*$ is one of the total functions (indeed, one of the *hypotheses*) that one can find inside $\mathcal{F}$ (where, $\mathcal{F}$ also includes partial functions that are also plausible hypotheses, but not plausible functions to work as the ground truth).
> > > > > > > > > > > >
> > > > > > > > > > > > I am just thinking out loud, trying to help. What do you think?

---

> > > > > > > > > > > > > ### Comment · Reviewer_mVCw · 2023-08-20
> > > > > > > > > > > > >
> > > > > > > > > > > > > Note also that if $\mathcal{F}$ is extended to include partial functions it can potentially affect some of the complexity parameters that you have, as the learner has to explore a larger search space in order to identify a good hypothesis.
> > > > > > > > > > > > >
> > > > > > > > > > > > > Having said that, the VC dimension which is discussed in the paper is _not_ affected because one *cannot* shatter a larger set of points when using the entire $\mathcal{F}$ compared to what one can shatter by looking only at the total functions that are found inside $\mathcal{F}$.  Is this true for the rest of the complexity parameters that you discuss/mention/comment upon in the paper? This needs to be checked by you.

---

> > > > > > > > > > > > > > ### Author Response · Authors · 2023-08-20
> > > > > > > > > > > > > >
> > > > > > > > > > > > > > This is a really good observation. In fact, all our results only depend on the complexity of the total functions which can be the labelings (called $\mathcal{F}$ in the above comment.

---

> > > > > > > > > > > > > > > ### Comment · Reviewer_mVCw · 2023-08-20
> > > > > > > > > > > > > > >
> > > > > > > > > > > > > > > Thank you for all the clarifications.  I am happy that we were able to communicate and be in agreement and that the final paper will be polished and present your work in an easier to understand way.
> > > > > > > > > > > > > > >
> > > > > > > > > > > > > > > I am looking forward to the final version of the paper.  I have increased the score from 3 (reject) to 6 (weak accept).
> > > > > > > > > > > > > > >
> > > > > > > > > > > > > > > Thank you for a very interesting paper!

---

> > > > > > > > > > > > > > > > ### Author Response · Authors · 2023-08-20
> > > > > > > > > > > > > > > >
> > > > > > > > > > > > > > > > We thank you again for the engaged discussion. This has been really valuable for us to improve the paper.

---

> > > > > > > > > > > > > ### Author Response · Authors · 2023-08-20
> > > > > > > > > > > > >
> > > > > > > > > > > > > We thank the reviewer again for engaging thoughtfully with our paper and we sincerely appreciate the effort.
> > > > > > > > > > > > >
> > > > > > > > > > > > > This way of thinking of the model is indeed extremely close to the model that we have been working with mentally and what we were trying to convey in our earlier responses. Just for clarity, in this perspective it is easiest to think of having 3 function classes. The first $\mathcal{F}$ which is the class from which potential labelling functions come from. These functions are all total and have no abstentions in them. The second is the class that you mention, let us call this $\tilde{\mathcal{F}}$ which is class constructed by taking $\mathcal{F}$ and adding all possible consistent partial functions. If one is considering a purely batch version of the problem we consider in our paper it is very reasonable to just consider these two classes (related model with just iid data have been considered in the literature, eg see https://arxiv.org/abs/2107.08444) and restrict to learner who just look for functions in $\tilde{\mathcal{F}}$.
> > > > > > > > > > > > >
> > > > > > > > > > > > > When working with online prediction (as we do in our paper), it usually is easiest to consider a third class (of partial functions), which we will call $\mathcal{G}$ in this discussion, which corresponds to the class of functions that the predictor is allowed to use to make predictions (this might be useful to consider in the batch setting as well). As you mentioned earlier in the discussion, the learners power can come from the fact that $\mathcal{F} \subset \mathcal{G}$. In our protocol, we impose no restrictions on $\mathcal{G}$ *a priori* (that said, that algorithm might choose to restrict its actions to a smaller subclass in order to work but this is not impose upon the algorithm in our model). That is, we allow $\mathcal{G}$ to be the set of all partial functions. This is in close correspondence with the traditional mistake bound model (Sec 21.1 here https://www.cs.huji.ac.il/~shais/UnderstandingMachineLearning/understanding-machine-learning-theory-algorithms.pdf) where the learner is allowed to output a predicted label $\hat{y_t}$ after looking at $x_t$ (with no consistency with any fixed function in the original labelling class required). In order to see that this sort of predictions outside the class is useful in the online model compare the CONSISTENT algorithm with HALVING algorithm from https://www.cs.huji.ac.il/~shais/UnderstandingMachineLearning/understanding-machine-learning-theory-algorithms.pdf . The halving algorithm uses a point wise majority of functions in the original labelling class $\mathcal{F}$ (which in general need not be in the class) and achieves a much better mistake bound than the CONSISTENT algorithm. The key point is that even though the algorithm is allowed to use arbitrary functions to predict the complexity (number of mistakes) depends only of the labelling class $\mathcal{F}$.
> > > > > > > > > > > > >
> > > > > > > > > > > > > The analoguous result for the abstention case is exactly what we show. That is, for any labelling class $\mathcal{F}$, we allow online predictions with abstentions using an arbitrary class $\mathcal{G}$ and show that the error in the online model depends only on the complexity (VC dimension) of the labelling class $\mathcal{F}$ (of total functions).
> > > > > > > > > > > > >
> > > > > > > > > > > > > But again we are really grateful for this discussion and we will definitely include clarifications based on this discussion in the revisions of our paper.
> > > > > > > > > > > > >
> > > > > > > > > > > > >
> > > > > > > > > > > > > PS: if you are curious as to why it is interesting to allow the learner to prediction using arbitrary functions (would the learner not just overfit?). The answer turns out to be that even using arbitrary functions to learn (against a simple labelling class) in fact guarantees "generalization" (this needs to interpreted carefully). This is due to the fact that though the learner is allowed to use arbitrary functions to predict since they are only "looking" at part of the data for each prediction, this implies the "final model" is simple (a posteriori). A version of this can be found in Exercise 5 of Chapter 21 https://www.cs.huji.ac.il/~shais/UnderstandingMachineLearning/understanding-machine-learning-theory-algorithms.pdf).

---

### Official Review · Reviewer_cSg3 · 2023-07-04

**Soundness:** 4 excellent
**Presentation:** 4 excellent
**Contribution:** 3 good
**Rating:** 6
**Confidence:** 3

**Summary:**

This paper proposes a sequential prediction setting in which an adversary injects adversarial examples with clean labels, and the algorithm is allowed to abstain from predicting. This setting lies between the stochastic and the fully adversarial settings, which are known to be characterized by the VC and Littlestone dimensions of the hypothesis class.

The paper mainly proves the following two learning performance guarantees stated in terms of the VC dimension. (The algorithm incurs a unit loss whenever it outputs an incorrect label, or it abstains on a clean data poiunt.)
- An $O(d^2\log T)$ upper bound on the number of mistakes, where $d$ is the VC dimension and $T$ is the time horizon. This result requires the access to the marginal data distribution.
- An $O(\sqrt{T})$ error bound for the $d = 1$ case, without access to marginal distributions.

**Strengths:**

This paper defines a new setting of sequential prediction that models the power of abstention against clean-label attacks (e.g., adversarial examples). I found the setting well-motivated and the technical material beautifully presented. The authors identified various future directions of work on this model as well as potential connection to the learning theory literature.

Despite the weaknesses discussed below, I (weakly) lean towards accepting this submission.

**Weaknesses:**

- The error bounds are likely to be rather loose in terms of both $d$ and $T$.
- The assumption on having *perfect* access to the marginal data distribution is arguably too strong. While the approach should still work if the access to $\mathcal{D}$ is replaced with (unlabeled) samples from $\mathcal{D}$, the sample complexity would likely be $T^{\Omega(d)}$; see questions below.

**Questions:**

Following up on the second point in the weakness part: To run Algorithm 1 using unlabeled examples from $\mathcal{D}$, it seems that we need an estimation of $\rho_k$ up to an $O(\alpha_k) = O(T^{-k})$ additive error, which in turn requires $T^{\Omega(d)}$ samples when $k = d$. Is there a more efficient way of using unlabeled examples, or would you conjecture that such a dependence is likely to be necessary?

**Limitations:**

This work is purely theoretical and the limitations are the assumptions that underlie the problem setting as well as the theoretical results. These were addressed in the "Broader Impact" section in the paper.

---

> ### Author Rebuttal · Authors · 2023-08-09
>
> We thank the reviewer for the thoughtful review. Here we address the questions/weaknesses pointed out by the reviewer.
>
>
> **Looseness of the bounds**: For the known distribution setting, this is only loose by a factor of $d$ compared to the fully stochastic setting with no adversarial/OOD data. Since our algorithm needs to also handle adversarial/OOD data, it is not clear if this can be matched. For the unknown marginal setting, we do not yet know if it is possible for general VC classes. Hence, we do not know what the regret is likely to be in these settings
>
>
> **Knowledge of distribution**: Yes, we agree that this assumption is very strong. However, even under this assumption, it is not obvious how to solve the problem since the algorithm needs to decide "in-sample" vs "out of sample" using only a single sample at test time and corrupted training samples. The only setting where we can remove the assumptions are the special cases of VC dimension 1 classes, and axis-aligned rectangles. Furthermore, our work follows a long tradition of work in the TCS and learning theory literature that construct learning algorithms under known marginal distribution. In fact, a considerable fraction of the literature on computational learning theory focuses on narrower assumptions such as the marginal known to be Gaussian or uniform on the Hypercube.
>
>
> **Efficient use of unlabelled samples**: The reviewer correctly points out that the current algorithm would require a very large number of samples $T^{O(d)}$ to be able to get accurate enough estimates for the algorithm to work. We do not yet know of a way to make use of the unlabelled samples more efficiently. Recall that given $n$ i.i.d. samples, we can construct ${n \choose k} = O(n^k)$, $k$-tuples, so there is some hope that we do not necessarily need $T^{O(d)}$ samples if we can show stronger concentration using dependent sampling. However, it is not clear to us if this is possible in general. Perhaps, at the cost of a worse dependence on $T$ (like in our unknown distribution setting) we could get away with less accurate approximations of the shattering probabilities.

---

> > ### Comment · Reviewer_cSg3 · 2023-08-14
> >
> > Thank you for answering my questions! I don't have follow-up questions, and my overall evaluation of the paper remains positive.

---

### Official Review · Reviewer_KALR · 2023-07-07

**Soundness:** 3 good
**Presentation:** 3 good
**Contribution:** 2 fair
**Rating:** 5
**Confidence:** 3

**Summary:**

The paper presents a new protocol for beyond-worst-case sequential prediction, incorporating the option of abstention. It introduces two main algorithms: the first achieves an error rate of $O(d^2 \log T)$ for classes with VC dimension $d$, while the second realizes an error of $O(\sqrt{T})$ for a specific instance of classes with VC dimension $1$.

**Strengths:**

1. An extensive literature search and insightful discussion of prior work spanning a variety of areas. This includes nuanced connection to topics such as beyond-worst-case sequential prediction, abstention-based learning, adversarially robust learning, adversarial examples, testable learning, etc.

2. A new protocol of beyond-worst case sequential prediction with the option of abstention.

**Weaknesses:**

1. Although the authors propose a new protocol, I struggle to identify novelty, as the primary difference appears to be the relatively straightforward inclusion of an additional abstention option.

2. While new algorithms have been introduced, they are specifically tailored to special classes, thereby limiting the significance of the paper. Why that special case of VC dimension 1 classes is important?

3. The proof techniques presented appear to lack significant novelty. What, then, is the underlying challenge?

4. The phrase "naturally be extended" has been referenced multiple times throughout the paper, such as on lines 147-148, 366-367, and 370-371. Firstly, these extensions are not immediately evident, and as such, a more detailed discussion would be beneficial. Secondly, if these extensions are indeed straightforward, including them in the paper could greatly augment its significance.

**Questions:**

See Weaknesses.

**Limitations:**

The authors adequately addressed the limitations.

---

> ### Author Rebuttal · Authors · 2023-08-09
>
> We thank the reviewer for their thoughtful review. Here we address the major concerns raised by the reviewer.
>
>
> **Novelty of the model**: The inclusion of abstention itself is not sufficient to get any guarantees in this setting. We need the correct notion of regret to accompany this. For instance, an algorithm could just abstain on the entire sequence, but such an algorithm would not be useful. On the other hand, from classical results we know that asking the algorithm to predict always would lead to very pessimistic bounds (could lead to infinite errors even when the VC dimension is 1). Our model balances these two settings to get a beyond-worst case guarantee where the algorithm is required to predict correctly on the i.i.d. examples while allowing free abstention on adversarial/ood examples. This allows us to design algorithms that get similar error bounds to the setting with only i.i.d. examples and non-adversarial examples. Previous models like KWIK could not get such guarantees because they didn't account for an underlying i.i.d. distribution which is natural in several real-world applications.
>
> Further, we propose a loss corresponding to our setting given abstention and misclassification. But, our setting is not merely a new loss minimization setting. Since we do not have access to whether the examples the learner has seen are adversarially injected or i.i.d. ($c_t$ is unknown), the algorithm cannot compute its own loss (in particular, the learner cannot compute the abstention error). So the algorithm needs to work without directly knowing its own total loss. This highlights another major challenge of our setting. We believe this  more realistically captures OOD and adversarial examples in the real world.
>
> **Focus on particular classes**: Our known distribution algorithm is not tailored to special classes and works for all function classes as long as they have bounded VC dimension (which is a necessary condition for learning even for iid data). Note that the unknown distribution case is significantly more challenging than this as we only have the examples in the past (which may be corrupted) to make the decision for the example at each time, and we do not have any information about what the actual distribution could look like. As noted in the paper, even for the simple class of thresholds, designing an algorithm that works is already non-trivial. Currently, we do not know if this is even possible for general VC classes. Therefore, towards addressing the general question of VC classes, we naturally considered the class of VC dimension 1 (for example, also considered in closely related work by Blum et. al. "Robust learning under clean-label attack." COLT 2021). Note that we also extend the results to axis-aligned rectangles which can have much higher VC dimension.
>
>
> **Novelty of proof technique**: To our knowledge, the proof techniques in the paper are novel. We consider both the algorithm design (keeping track of the higher order shattering) and the analysis (the potential argument analyzing the tradeoff between abstentions and mistakes) as contributions of the paper. We do not know any other paper that uses these. Our algorithm in the known distribution setting is inspired by Hanneke's algorithm for active learning, however, the setting and our analysis differs significantly from theirs. The main challenge in our setting is to identify at each time step whether the example we see is in-distribution or adversarial/OOD. This is an extremely challenging problem given that our sequence so far could also have adversarial corruptions. It is not clear at all if this is even possible. Our algorithms create uncertainty measures by exploiting the structure of the underlying function class and the distribution in a highly non-trivial way, which we believe could have broader impact as a technique for uncertainty quantification for learning. **_Could the reviewer expand on the lack of novelty by pointing us to other works that use similar techniques?_**
>
>
> **Use of "naturally be extended to""**: We apologize for using this phrase lightly, and will add details of the extensions for each case. Most of the usage of this phrase refers to the model we propose and not the algorithms/proofs. Here we explain these briefly:
> - naturally be extended to the agnostic setting with adversarial/noisy labels
>   - To extend the model to the agnostic setting, we would need to modify the regret to include an additional error term to compare with the best possible error any predictor in the function class could get for the sequence.
>   - To extend to adversarial labels, we just allow the adversary to output any $y_t$
> - naturally be extended to more general forms of prediction such as multiclass classification, partial concept classes, and regression
>   - To extend the model to multiclass classification, we would need to modify the prediction to allow for predicting from a set of $k$-labels and the misclassification loss would need to be modified to penalize error if the predicted label is not the true label
>   - Similarly for regression, we would need to allow prediction to be real-valued and the misclassification loss to be squared loss or the absolute loss.
>   - To extend to partial concept classes, we can keep the model essentially the same with the constraint that the adversary can only inject examples that are consistent with a hidden partial concept, that is, cannot provide an example where the partial concept does not have a label. With this definition, the model allows for more general concepts such as halfspaces with margin while allowing for similar analysis and algorithms.

---

> > ### Comment · Reviewer_KALR · 2023-08-17
> >
> > Thank you for the response. While I recognize the model's novelty, I maintain my stance that the results for a specific instance of classes with a VC dimension of 1 are constrained, and the primary technique employed seems limited in scope. Consequently, my score remains unchanged.

---

> > > ### Author Response · Authors · 2023-08-17
> > >
> > > Thank you for your response. We understand your stance but we wanted to mention something brief regarding why VC one is very interesting theoretically.
> > >
> > > VC one class include thresholds and the indicators of points. Though both these classes are simple, they have large Littlestone dimension (complexity of sequential prediction) and star numbers (complexity of active learning) respectively. Thus, an upper bound for all VC one classes rules out  lower bounds purely in terms these other candidate complexity measures. This is particularly interesting for the case of the Littlestone dimension since it is the complexity for sequential prediction. Thus our VC one upper bound shows that these measures for closely related models *do not* capture the complexity in our model. and one can genuinely hope that there is a universal bound in terms of the VC dimension.

---

### Official Review · Reviewer_7wBe · 2023-07-13

**Soundness:** 3 good
**Presentation:** 3 good
**Contribution:** 2 fair
**Rating:** 5
**Confidence:** 3

**Summary:**

This paper proposes a pipeline and algorithms for machine learning prediction with abstention. The authors first propose the optimization framework to learn a model which allows abstention. They then consider different distributions and propose different algorithms for the learning process. Their theoretical analysis shows that the algorithms can achieve low error rate for misclassification and abstention.

**Strengths:**

* The paper studies a variety of different contributions. The authors start with one-dimensional threshold scenario and shows the application of the abstention model. Then they provide algorithms for both known distribution and unknown distribution. The latter one is a more difficult setting and the authors provide a generalized algorithm for it based on the former one.

* The paper provides comprehensive error bound for the algorithms. I am not an expert in theoretical ML analysis, but as far as I can tell, the theory part does not contain significant mistakes.

**Weaknesses:**

* I do not see significant real-world application of the proposed abstention framework with adversarial resilience. The authors mention the case of predicting patient's illness, in which case I understand why the abstention would be necessary. However, I do not see the necessity of caring about the adversarial resilience in such case - is it a (serious) threat that an adversary may launch a clean-label attack when people are training an illness classification task?

* The authors also emphasize the "sequential prediction" in addition to the abstention framework and adversarial resilience in the context. I am not fully clear why this is also an important point to consider - can we also apply the proposed pipeline on a static dataset that does not contain a sequential order but also benefits from the abstention pipeline?

* The paper does not contain empirical evaluation of the proposed algorithm, nor does it contain comparison with other works to show how good their bound is.

**Questions:**

* What will be the practical use case for the abstention framework under the adversarial setting?

* Can the algorithm also be applied to get an abstention model for a non-sequential dataset?

**Limitations:**

The authors discuss their limitation in their broader impact setup. The paper does not contain potential negative societal impact.

---

> ### Author Rebuttal · Authors · 2023-08-09
>
> We thank the reviewer for their thoughtful review. Here we address the major concerns raised by the reviewer.
>
>
> **Significance of the model in the real-world**:  The key benefit of our model and algorithms is the fact that they produce 'certain' predictions. In particular, the model is correct whenever it predicts despite out-of-distribution or adversarial data. Creating predictors equipped with this guarantee is very desirable in high-stakes applications where incorrect predictions are costly, and we need high-levels of reliability from our models. As for the medical scenario, the out-of-distribution setting is more natural to think about. For example, if our model has seen data mostly from a particular population (say the majority population), then our model is likely to make incorrect predictions on patients not from this population. In such scenarios, it is better for the predictor to abstain than make an incorrect prediction on the minority population. Similar out-of-distribution interactions can occur in other settings such as self-driving cars, chatbots, etc.
>
> Further, we propose a loss corresponding to our setting given abstention and misclassification. But, our setting is not merely a new loss minimization setting. Since we do not have access to whether the examples the learner has seen are adversarially injected or i.i.d. ($c_t$ is unknown), the algorithm cannot compute its own loss (in particular, the learner cannot compute the abstention error). So the algorithm needs to work without directly knowing its own total loss. This highlights another major challenge of our setting. We believe this more realistically captures OOD and adversarial examples in the real world.
>
>
> **Importance of considering sequential prediction**: Most real-world settings involve interactions with the model, and dynamic decision making, where the environment can change over time. Therefore, it is natural to consider prediction in the sequential setting. On the technical side, this setting is also much more challenging than the offline setting. In our case, the sequential part introduces several challenges:
> - we need to decide to predict or abstain based on only one example
> - the adversary can change their strategy over time adaptively based on our decisions.
>
> That said, our techniques can be directly applied to a static dataset to achieve similar guarantees. But due to the relative simplicity of the static setting, more direct approaches have been previously studied as described in our related work section under the title “abstension-based learning”.
>
>
> **Comparison to other bounds**: In this work we have proposed a new model, therefore a direct comparison to existing bounds is not possible. As described in the paper, the algorithms designed for the stochastic or worst-case settings do not get non-trivial guarantees in our model. In particular, prior to our work, it was not even clear if we could get any rates better than the Littlestone dimension (which can be infinite even when the VC dimension is finite) in the non-stochastic setting with arbitrarily many adversarial injections. We refer the reviewer to our extensive related work section that compares our results to closely related other work.
>
>
> **Empirical evaluation**: The main focus of our work is mainly theoretical and we believe empirical evaluation distracts from the main message of the paper. Our goal in this paper was to propose a model under which sets a realistic goal on handling adversarial/OOD data and showing the existence of efficient algorithms in this setting. We hope that our work will inspire practitioners to develop practical algorithms using the algorithmic ideas proposed in this work. One particular direction that would be great to explore, would be to use our uncertainty estimation measures based on shattered sets to create heuristics that work for more complex classes such as neural networks.
>
>
> **Practical use of abstention**:  Currently the proposed algorithms are not efficient, however there is hope to create heuristics based on our approaches that could measure the model's uncertainty which would have great practical usage. Even in the setting of adversarial examples (as in vision), equipping the model with abstention would allow us to prevent adversarial attacks which are outside of the perturbation sets we have trained for. Our related work highlights some ways in which this has been used in practice in the static setting.

---

> > ### Comment · Reviewer_7wBe · 2023-08-17
> >
> > Thanks for the clarification and explanation. Although I am still concerned with the practicalness of the proposed algorithm, I agree with the authors that the considered sequential prediction scenario is an important one to study on and the proposed framework is a good theoretical guide. Therefore, I would raise my final score to 5.

---

### Decision · Program_Chairs · 2023-09-21

**Decision:**

Accept (poster)

**Comment:**

The paper presents a sequential prediction problem that allows the option of abstention, where an attacker can inject clean-label adversarial examples. It introduces two algorithms with associated error bounds; in one the error scales with the VC dimension of the hypothesis class. The reviews are all on the positive side, if not all strongly, after a very healthy amount of discussion and response.
.